# Partial Ring Scan: Revisiting Scan Order in Vision State Space Models

Yi-Kuan Hsieh [1]   Kuan-Chuan Peng [2]   Xin Li [3]   Ming-Ching Chang [4]   Yu-Chee Tseng [1]   Jun-Wei Hsieh [1] *

## Abstract

State Space Models (SSMs) provide linear-time alternatives to attention for vision, but require serializing 2D images into 1D sequences using a predefined scan order. We identify scan order as a previously underexplored inductive bias that fundamentally shapes spatial dependency modeling in Vision SSMs. Fixed scan paths distort local adjacency, fragment object structure, and induce anisotropic representations that are brittle under geometric transformations such as rotation. We propose Partial RIng Scan Mamba (PRIS-Mamba), a rotation-robust traversal that decomposes images into concentric rings, performs permutation-invariant aggregation within each ring, and models cross-ring dependencies via short radial SSMs. This design induces a structured factorization of spatial dependencies that preserves isotropy while maintaining linear complexity. To improve efficiency without sacrificing expressivity, we introduce partial channel filtering, selectively applying recurrent modeling to informative channels while routing others through a residual pathway. Empirically, PRIS-Mamba improves accuracy, efficiency, and rotation robustness over prior Vision SSMs on ImageNet-1K. Our results position scan-order design as a core representational choice in Vision SSMs, with implications for robustness and generalization beyond architectural scaling. The code will be released upon paper acceptance.

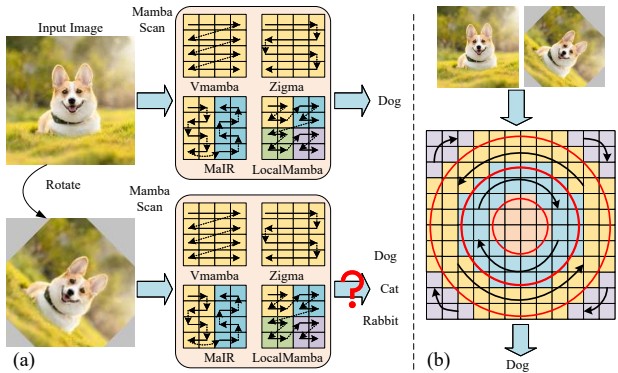

*Figure 1.* **Scanning order affects Vision-Mamba performance.** (a) *Fixed-path scans* (*e.g*, raster or serpentine in VMamba (Liu et al., 2024), Zigma (Hu et al., 2024), MaIR (Li et al., 2025), LocalMamba (Huang et al., 2024)) preserve sequence–space alignment only under flips. An rotation causes padding and global reindexing, fracturing the path so the recurrent kernel moves along misaligned neighborhoods. (b) *Our Ring Scan* treats serialization as *order-agnostic* aggregation within *concentric rings*, followed by *radial* composition from inner to outer, producing a rotation-stable sequence without polar remapping or rotation-specific training.

## 1. Introduction

Recurrent State Space Models (SSMs) have recently emerged as a competitive alternative to attention-based architectures for long-context vision, offering linear-time sequence processing while maintaining strong accuracy (Gu et al., 2021; Gu & Dao, 2024). Vision adaptations such as Vision Mamba, VMamba (Liu et al., 2024), and PlainMamba embed selective SSM blocks into hierarchical backbones, which requires *serializing* a 2D image into a 1D token sequence according to a chosen **scan order** (Zhu et al., 2024a; Liu et al., 2024; Yang et al., 2024). Although the scan order is often treated as a mere implementation detail, *e.g*, using row-wise, column-wise, or serpentine traversals, it implicitly defines which spatial neighbors appear adjacent in the sequence. This adjacency governs what local structures the SSM can effectively model with short-range recurrences.

Mounting evidence shows that the scan order is far from innocuous. Large-scale studies demonstrate that patch ordering can produce statistically significant performance differences, sometimes exceeding ten Dice points in segmentation tasks (Hardan et al., 2025). In remote sensing, experiments

[1]College of Artificial Intelligence, National Yang Ming Chiao Tung University, Taiwan (R.O.C.), Tainan [2]Mitsubishi Electric Research Labs, U.S.A [3]Department of Computer Science, State University of New York at Albany, U.S.A, New York [4]Department of Electrical & Computer Engineering, State University of New York at Albany, U.S.A, New York. Correspondence to: Jun-Wei, Hsieh <jwhsieh@nycu.edu.tw>.

*Proceedings of the 43rd International Conference on Machine Learning*, Seoul, South Korea. PMLR 306, 2026. Copyright 2026 by the author(s).

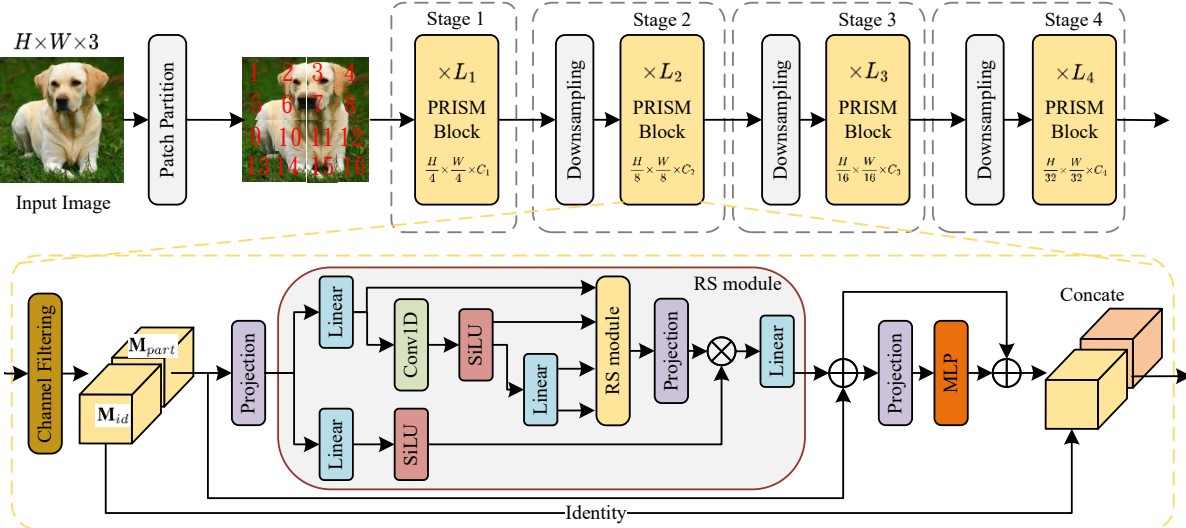

*Figure 2.* **Architecture with Partial RIng Scan Mamba (PRIS-Mamba).** The image is patchified and processed by a four-stage backbone; stage $i$ stacks $L_i$ **PRISM** blocks (*Partial RIng Scan Mamba*) with $C_i$ output channels, and stages are separated by downsampling. Each PRISM performs order-agnostic aggregation over a *subset of concentric rings* (partial ring scan), composes information radially with a short sequence operator, and writes features back via a $1\times1$ projection before residual fusion. Channel filtering routes only the most informative channels while keeping the rest on a lightweight residual branch for further efficiency improvement.

indicate that complex multi-directional scans do not consistently outperform simpler rasterizations, which can suffice depending on the domain (Zhu et al., 2024b). These observations suggest that the scan order should be treated as a first-class, cost-free hyperparameter that mediates the alignment between *sequence adjacency* (what the SSM processes) and *geometric adjacency* (the true image structure).

In this paper, we examine the role of the scan order in modern Vision State Space Models (SSMs) and use this analysis to motivate a new traversal and architectural redesign. Conventional SSM traversal patterns often disrupt object continuity: when objects extend in directions misaligned with the scan path, their patches become interleaved with unrelated regions, forcing the SSM to spend capacity repairing local coherence rather than modeling semantics. The problem worsens under geometric transformations such as in-plane rotations, which globally reindex the image and amplify discontinuities. We show that the scan order can affect performance by altering spatial adjacency, fragmenting object structure, and increasing sensitivity to distortions.

To address this, we propose **Partial RIng Scan Mamba (PRIS-Mamba)**, a rotation-robust traversal that decomposes an image into concentric rings, forms compact ring-level representations, and propagates context radially using a selective SSM. Fig. 1 illustrates that conventional scan paths (*e.g*, raster, bidirectional, diagonal, serpentine) are brittle under rotation, as they disrupt token adjacency and fragment object structure. In contrast, the proposed ring scan aggregates features within each ring and processes ring descriptors from inner to outer, preserving spatial locality without relying on angle-specific ordering. This construc-

tion is an intuitive way to preserve neighborhood structure under rotation without doing polar remapping or rotation-specific training. Moreover, When combined with detectors such as YOLO (Jocher, 2024; Tian et al., 2025), ring scan naturally extends to object-aware SSMs, further aligning traversal structure with object geometry.

We evaluate across classification, detection, and segmentation tasks. On ImageNet-1K (224×224), PRIS-Mamba achieves 84.5% Top-1 with 3.9G FLOPs and 3,054 img/s on A100 GPU, outperforming VMamba (Liu et al., 2024) (82.6%, 5.6G, 1,686 img/s) by 1.9% while using 30% fewer FLOPs and roughly 1.5× higher throughput. A PRIS-Mamba variant without channel filtering attains 84.1% at 4.6G and 2,177 img/s, showing that the *Partial* module reduces FLOPs and improves accuracy. On MS COCO (1× schedule, 1280×800), PRIS-Mamba attains 48.9 AP[box] and 43.2 AP[mask] with 235G FLOPs, outperforming VMamba (46.5/42.1 at 262G) and GroupMamba (47.6/42.9 at 279G) while using 10–19% less FLOPs. Because the token count is unchanged and recurrences are short for each ring, memory and runtime remain close to standard Vision-SSMs while delivering clear gains in accuracy and rotation robustness.

Our main contributions are summarized as follows:

- **SSM Scan-order Analysis:** We provide the first systematic study of how traversal paths shape spatial adjacency in Vision SSMs. We show that common raster and serpentine scans can fracture object continuity and are highly sensitive to geometric transformations.
- **Ring Scan for Vision SSMs:** Building on this analysis, we introduce a rotation-stable scanning scheme that

groups pixels into concentric rings, performs order-agnostic aggregation within each ring, and propagates information radially through short selective SSMs. This preserves the linear-time SSM core while avoiding the fragility of global path-based serialization.

- **Partial Channel Filtering (PCF):** We enhance efficiency by forwarding only the most informative channels through the ring-wise recurrent path and routing the remainder through a lightweight residual branch. This improves throughput and reduces FLOPs while modestly improving accuracy.

- **Unified architecture and empirical results:** Integrating the above ideas yields PRIS-Mamba, achieving the state-of-the-art performance among Vision SSMs. It improves accuracy and throughput on ImageNet-1K and COCO while using fewer FLOPs, and it substantially increases robustness to rotation without any rotation-specific training.

## 2. Related Work

### 2.1. Convolutional Neural Networks (CNNs)

CNNs have been the cornerstone of visual recognition since AlexNet (Krizhevsky et al., 2012). Extensive research has continuously advanced their modeling power (Simonyan & Zisserman, 2014; Szegedy et al., 2015; He et al., 2016; Huang et al., 2017) and computational efficiency (Howard et al., 2017; Tan & Le, 2019; Yang et al., 2021; Radosavovic et al., 2020) across diverse vision tasks. Advanced operators such as depthwise (Howard et al., 2017) and deformable convolutions (Dai et al., 2017; Zhu et al., 2019) further improved flexibility and representational capacity.

More recent efforts, inspired by the success of Transformers (Vaswani et al., 2017), modern CNNs (Liu et al., 2022b) incorporate long-range dependencies (Ding et al., 2022b; Rao et al., 2022; Liu et al., 2022a) and dynamic weighting mechanisms (Han et al., 2021). These hybrid architectures achieve strong accuracy while preserving the inductive biases and efficiency advantages of convolution.

### 2.2. Vision Transformer (ViTs)

The Vision Transformer (ViT) (Dosovitskiy et al., 2020) first showed that a pure Transformer architecture can achieve strong performance on visual recognition, highlighting the importance of large-scale pre-training. To mitigate ViT's reliance on massive datasets, DeiT (Touvron et al., 2021) introduced a teacher–student distillation strategy that transfers CNN inductive biases to ViTs. Building on this foundation, numerous works proposed hierarchical and locally aware variants (Liu et al., 2021; Dong et al., 2022; Wang et al., 2021; gao et al., 2021; Zhang et al., 2023; Tian et al., 2023; Dai et al., 2021; Ding et al., 2022a; Zhao et al., 2022; Ali

et al., 2021) that improve scalability and efficiency.

Another major research direction aims to alleviate the quadratic complexity of self-attention. Linear Attention (Katharopoulos et al., 2020) reformulates attention as a linear dot product of kernel feature maps, reducing computational cost from quadratic to linear. GLA (Yang et al., 2023) introduces a hardware-friendly design that balances memory access and parallelism. RWKV (Peng et al., 2023) integrates linear attention with RNN-style inference, enabling parallelizable training while preserving recurrent efficiency. RetNet (Sun et al., 2023) incorporates gating to construct a fully parallelizable alternative to recurrence, while RMT (Fan et al., 2024) extends this paradigm to vision by adapting temporal decay mechanisms into the spatial domain for representation learning.

Recent ViT research has aimed to improve computational efficiency and better capture spatial dependencies. SHViT (Yun & Ro, 2024) integrates single-head self-attention with convolutional layers to reduce redundancy in early stages, achieving faster inference and higher accuracy on GPUs and mobile devices. GCViT (Hatamizadeh et al., 2023) combines global and local attention to handle multi-scale spatial interactions, yielding strong results in classification and segmentation. Scale-Aware Modulation Transformer (SMT) (Lin et al., 2023) employs multi-head mixed convolution and scale-aware aggregation to model the transition from shallow to deep dependencies, achieving notable accuracy improvements. TransNeXt (Shi, 2024) introduces biomimetic foveal attention for efficient visual processing and information fusion with fewer parameters.

### 2.3. State Space Models (SSMs)

While Vision Transformers achieve strong performance, their quadratic attention cost limits scalability to high-resolution inputs. State Space Models (SSMs) have emerged as efficient alternatives, offering linear-time sequence modeling with strong long-range dependency capture (Dao et al., 2022; Dao, 2023; Peng et al., 2023; Sun et al., 2023; Ma et al., 2022). HiPPO initialization (Gu et al., 2020) enables SSMs to model long sequences effectively, and the S4 framework (Gu et al., 2021) improves efficiency through normalized diagonal parameterization. Building on this, structured variants have been proposed, including complex-diagonal parameterizations (Gupta et al., 2022; Gu et al., 2022), multi-input multi-output extensions (Smith et al., 2022), diagonal-plus-low-rank decompositions (Hasani et al., 2022), and adaptive selection mechanisms (Gu & Dao, 2024), which have been incorporated into large-scale models (Mehta et al., 2022; Ma et al., 2022; Fu et al., 2022).

Although SSMs have excelled in text and speech, their use in vision remains underexplored. Recent work begins to close this gap by adapting Mamba-style SSMs to images,

balancing linear-time sequence modeling with 2D inductive biases in scanning, frequency, and architecture. Efficiency-focused designs include *Adventurer* (Wang et al., 2025b), which optimizes Vision-Mamba backbones for faster training without accuracy loss, and *TinyViM*, which uses hybrid Conv–Mamba blocks to capture low-frequency content via a Laplace-domain mixer (Wang et al., 2025b; Ma et al., 2025).

Scan-path design is a key focus. *PlainMamba* uses continuous 2D scanning with direction-aware updates for non-hierarchical recognition (Yang et al., 2024); *FractalMamba* employs fractal curves to better preserve multi-scale neighborhoods (Xiao et al., 2025); *ZigMa* applies DiT-style zigzag scanning to diffusion, improving speed and memory at high resolution (Hu et al., 2024). Beyond fixed paths, *Def-Mamba* learns deformable, content-adaptive scans to prioritize salient structures (Liu et al., 2025a), while *VSSD* leverages bidirectional visual context via non-causal SSMs (Shi et al., 2025). For restoration, *MaIR* enforces locality and continuity using nested S-shaped routes with lightweight fusion, achieving the state-of-the-art results on structure-sensitive benchmarks (Li et al., 2025).

Other refinements include *Mamba-Reg*, which inserts evenly spaced register tokens to stabilize scaling and suppress high-norm background artifacts (Wang et al., 2025a). Task-specific extensions demonstrate Mamba's versatility: *Selective Visual Prompting* streamlines adaptation of Vision-Mamba backbones, and *Mamba as a Bridge* fuses foundation vision and vision-language priors for domain-generalized semantic segmentation, achieving competitive mIoU (Yao et al., 2025; Zhang & Tan, 2025).

### 2.4. Scan-Based Vision SSMs

Recent Vision Mamba architectures adapt selective state-space blocks to visual data by *scanning* 2D feature grids to define a 1D token processing order, along with the SSM propagates states. VMamba (Liu et al., 2024) integrates a 2D Selective Scan (SS2D) into a hierarchical backbone, achieving strong accuracy–efficiency trade-offs across both classification and dense prediction tasks (Liu et al., 2024). For medical and remote-sensing segmentation, VM-UNet replaces or augments attention with Visual State-Space blocks in a U-shaped design, improving long-range dependency modeling at low computational cost (Ruan et al., 2024).

Beyond single-path rasterization, 2DMamba explicitly incorporates 2D spatial structure through selective operators, improving representation quality on both natural and whole-slide imagery (Zhang et al., 2025). Recent surveys (Zhang et al., 2024; Liu et al., 2025b) systematically categorize these Vision Mamba variants, identifying emerging design patterns such as multi-directional or axial scans, tiled or windowed scanning, and lightweight positional priors. Hardware-aware adaptations further streamline selective scanning for mobile or edge deployment (Pei et al., 2025).

Despite this progress, most pipelines still rely on *flip-only* augmentation to preserve scan continuity. When inputs undergo arbitrary rotations, fixed-path traversals suffer from boundary padding and reindexing effects that disrupt token adjacency, which highlights the need for traversal schemes that remain geometrically consistent under rotation.

## 3. Method

### 3.1. From SSMs to Vision SSMs

Let $\{x_k\}_{k=1}^T$ be a token sequence with token width $m \in \mathbb{N}$ and length $T \in \mathbb{N}$. A small projector $\Pi(x_k)$ produces stepwise parameters $A_k, B_k \in \mathbb{R}^d$ (diagonal) and a mixing matrix $C_k \in \mathbb{R}^{d \times d}$. With $\odot$ denoting the Hadamard product, a selective SSM maintains a latent state $h_k \in \mathbb{R}^d$ and output $y_k \in \mathbb{R}^d$ as:

$$h_k = A_k \odot h_{k-1} + B_k \odot x_k, \ y_k = C_k h_k, \quad (1)$$

and $h_0 = 0$. Vision SSMs (VSSMs) integrate this sequence operator into a hierarchical backbones by first *serializing* a 2D feature map $X \in \mathbb{R}^{H \times W \times C}$ into a 1D sequence according to a chosen *scan order*. After the SSM processes the sequence, the results are written back to the grid through a $1 \times 1$ projection. The scan order therefore becomes a key design choice, since it determines which spatial neighbors are treated as adjacent in sequence space.

### 3.2. Scan Orders in Vision SSMs

A scan order assigns each grid location $(u, v)$ a unique visit index $k \in \{1, \dots, N\}$, where $N = HW$, thereby linearizing the grid. Fig. 6 illustrates twelve primitive orders (*e.g*, raster, serpentine, diagonal) and their composites that we use to study scan effects. These continuous paths generally perform well under horizontal or vertical flips, as sequence adjacency aligns with spatial neighbors. However, in-plane rotations permute neighborhood relations and introduce padded corners, breaking the correspondence between the 1D dynamics of Eq. (1) and the underlying 2D structure. This motivates a traversal that maintains stable groupings under rotation while keeping sequences short to preserve linear-time efficiency.

### 3.3. Ring-by-Ring Alternating Scan

We group pixels by their Euclidean distance to the image center $(c_x, c_y)$ and visit the grid *ring by ring* as in Fig. 3. For a pixel $(u, v)$ with $u \in \{0, \dots, W-1\}$ and $v \in \{0, \dots, H-1\}$, define its radius $r(u, v) = \left\| (u - c_x, \ v - c_y) \right\|_2$. Given a ring width $\Delta r > 0$, assign the integer ring index:

$$\hat{r}(u, v) = \left\lfloor \frac{r(u, v)}{\Delta r} \right\rfloor, \ \mathcal{P}_r = \{(u, v) : \hat{r}(u, v) = r\}. \quad (2)$$

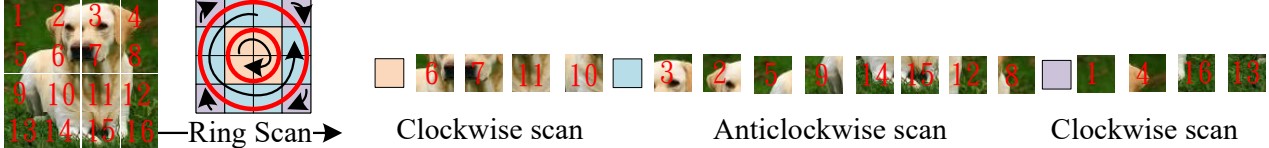

*Figure 3.* **Ring Scan.** Pixels are partitioned into concentric rings, which are interactively traversed in a clockwise or counterclockwise sequence. The resulting features are then aggregated in an order-independent fashion, proceeding from inner to outer rings.

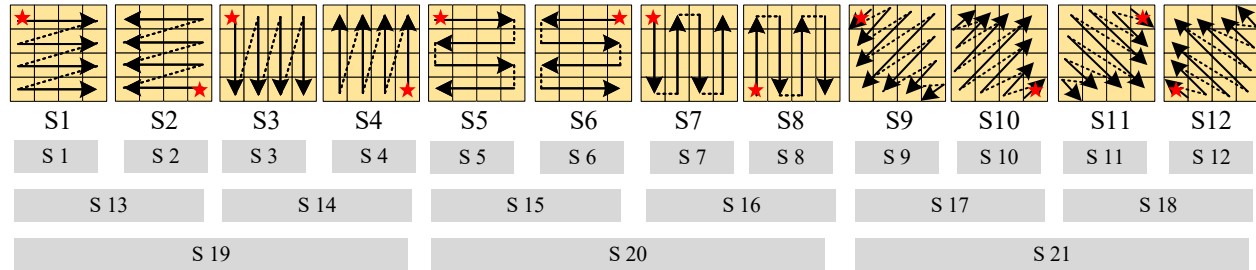

*Figure 4.* **Primitive scanning orders.** Twelve canonical paths (S1–S12) such as left-to-right raster, serpentine, and diagonal produce distinct 1D sequences from the same image. S1-12 evaluate single scans; S13–18 evaluate pairs of scans; S 19–21 aggregate four scans, enabling a systematic comparison of the effects of the scanning orders.

In real implementations, if an object detector is performed first, $(c_x, c_y)$ can be an object's center to make our method an object-aware descriptor. Ablation studies on the choice of $\Delta r$ were analyzed in Section D of the supplement.

**Alternating loop order:** Within each ring $r$, we construct a simple closed loop $\sigma_r : \{1, \ldots, L_r\} \to \mathcal{P}_r$ (any fixed contour rule works) and *alternate the traversal direction across rings*: clockwise for odd $r$ and counterclockwise for even $r$. Let $P \in \mathbb{R}^{m \times C}$ be a $1 \times 1$ projection. The ring-$r$ token sequence is:

$$x_{r,k} = P X_{\sigma_r(k)} \in \mathbb{R}^m, \qquad k = 1, \ldots, L_r, \quad (3)$$

processed in the chosen direction. A lightweight selective SSM runs along the loop:

$$
\begin{aligned}
h_{r,k} &= A_{r,k} \odot h_{r,k-1} + B_{r,k} \odot x_{r,k}, \\
y_{r,k} &= C_{r,k} h_{r,k}, \qquad h_{r,0} = 0,
\end{aligned}
\quad (4)
$$

and the ring descriptor is the average of all per-step outputs:

$$z_r = \frac{1}{L_r} \sum_{k=1}^{L_r} y_{r,k} \in \mathbb{R}^d. \quad (5)$$

**Rotation behavior:** In-plane rotations leave each pixel's ring index $\hat{r}(u, v)$ unchanged. As a result, a ring's loop experiences only a cyclic shift, and alternating traversal directions across rings reduce start-index bias. This design achieves rotation robustness without relying on polar remapping or rotation-specific augmentation.

### 3.4. Inner to Outer Radial Ring Scan

The per-ring descriptors $\{z_r\}_{r=0}^{R^\star}$ form a short radial sequence from the innermost to the outermost ring. Context is propagated inward-to-outward using a second selective SSM:

$$
\begin{aligned}
h_r^{\mathrm{rad}} &= \tilde{A}_r \odot h_{r-1}^{\mathrm{rad}} + \tilde{B}_r \odot z_r, \\
y_r^{\mathrm{rad}} &= \tilde{C}_r h_r^{\mathrm{rad}}, \qquad h_{-1}^{\mathrm{rad}} = 0.
\end{aligned}
\quad (6)
$$

This two-level design, consisting of a per-ring loop in Eq. (4) and a short radial chain in Eq. (6), maintains linear-time complexity: $O(L_r)$ per ring and $O(R^\star + 1)$ across rings. With practical choices of $\Delta r$, the number of rings $R^\star \ll \sqrt{HW}$, making the additional recurrence cost negligible compared to $O(HW)$ for scanning the entire grid.

### 3.5. Partial Channel Filtering

Before ring processing, we introduce a lightweight *hard routing* over channels to curb the per-channel cost of Ring Scan Module. Feature channels often exhibit unequal salience—many are weak or redundant yet would still incur full recurrent updates if all channels were forwarded. Unlike channel-attention methods such as SE (Hu et al., 2018) and CBAM (Woo et al., 2018), which learn *soft* importance weights but continue to process *all* channels through downstream blocks, our filter quickly separates informative channels from residual ones and only the informative subset enters the ring pathway. This maintains accuracy while substantially reducing FLOPs and activation memory.

Concretely, given a feature map $\mathbf{X} \in \mathbb{R}^{H \times W \times D_c}$ with channels $\{C_i\}_{i=1}^{D_c}$, we compute per-channel salience via global average pooling (GAP), $\mu_i = \mathrm{GAP}(C_i)$, $i = 1, \ldots, D_c$, and set a mean threshold: $\mu = \frac{1}{D_c} \sum_{i=1}^{D_c} |\mu_i|$. A channel is retained if $\mu_i \geq \mu$ and otherwise routed to a residual bypass. Denoting the partition operator by $\Phi$, we obtain:

$$(\mathbf{M}_{\mathrm{part}}, \mathbf{M}_{\mathrm{id}}) = \Phi(\mathbf{X}, \mu), \quad (7)$$

where $\mathbf{M}_{\text{part}} = \{ C_i \mid \mu_i \geq \mu \}, \mathbf{M}_{\text{id}} = \{ C_i \mid \mu_i < \mu \}$. The informative branch $\mathbf{M}_{\text{part}}$ is fed to the ring-by-ring modules, while $\mathbf{M}_{\text{id}}$ follows an identity/residual path and is fused later. Choosing the mean yields $O(D_c)$ complexity and avoids sorting; a median variant is possible but costs $O(D_c \log D_c)$. This fast selection preserves accuracy and achieves clear computational savings compared to soft reweighting approaches (Hu et al., 2018; Woo et al., 2018).

### 3.6. Write-Back and Fusion

Let $\Psi \in \mathbb{R}^{C \times d}$ be a $1 \times 1$ projection. For pixel $(u, v)$ on ring $\hat{r}(u, v)$, we broadcast the ring-level output:

$$Y_{u,v} = \Psi \, y_{\hat{r}(u,v)}^{\text{rad}} \in \mathbb{R}^C. \tag{8}$$

To avoid vanishing gradient issues, we fuse with the backbone stream using a lightweight residual projection:

$$X_{\text{out}} = X_{\text{in}} + \text{Conv}_{1 \times 1}(Y), \tag{9}$$

which preserves spatial size $H \times W$ and channel size $C$.

## 4. Experiment Results

**Implementation details:** Our models were trained from scratch using the AdamW optimizer (Loshchilov & Hutter, 2017) over 300 epochs with a batch size 128. The training regime included a linear warm-up during the first 5 epochs, a momentum of 0.9, and a cosine learning rate schedule (Loshchilov & Hutter, 2016) with an initial learning rate $1 \times 10^{-4}$. We adopted the same data augmentation techniques as prior work (Touvron et al., 2021), including mixup (Zhang et al., 2017), random erasing (Zhong et al., 2020), and auto-augmentation (Cubuk et al., 2019). Throughput was measured on an Nvidia A100 GPU. We compare against 15 Vision-Mamba families listed in Table 1. The code for the ring scan implementation is provided in the supplement.

### 4.1. ImageNet-1K Results and Efficiency Analysis

Table 1 compares PRIS-Mamba variants with recent Vision-Mamba models at $224 \times 224$ resolution and ablates partial channel filtering (PCF) by contrasting PRIS-Mamba (w/o PCF) with the full model. Efficient ring traversal is enabled by a precomputed index table that allows direct memory access to each patch. Compared with strong baselines of similar computational cost (e.g, VMamba: 5.6G FLOPs, 82.6% Top-1; SparX-Mamba: 5.2G FLOPs, 83.5%), the proposed ring scan and PCF shift the accuracy–efficiency frontier. In particular, PRIS-Mamba (w/o PCF) attains 84.1% Top-1 with 4.6G FLOPs and 2,177 img/s, outperforming VMamba by +1.5 points while using approximately 18% fewer FLOPs and achieving higher throughput. We provide ablations on other scanning orders (Fig. 6) in the supplement.

*Table 1.* Performance comparison on ImageNet-1K all images are of size $224 \times 224$. Throughputs (TP) are measured on Nvidia A100 GPU.

| Model | Params | GFlops | TP(img/s) | Top-1 Acc% |
|---|---|---|---|---|
| Vim (Zhu et al., 2024a) | 26 M | 5.3 G | 811 | 80.5 |
| VMamba (Liu et al., 2024) | 30 M | 5.6 G | 1686 | 82.6 |
| SiMBA (Patro & Agneeswaran, 2024) | 27 M | 5.0 G | - | 84.0 |
| Zigma (Hu et al., 2024) | 31 M | 5.1 G | - | 82.4 |
| QuadMamba (Xie et al., 2024) | 31 M | 5.5 G | 1252 | 81.4 |
| LocalMamba (Huang et al., 2024) | 26 M | 5.7 G | - | 82.7 |
| FractalMamba (Xiao et al., 2025) | 31 M | 4.8 G | - | 83.0 |
| Adventurer (Wang et al., 2025b) | 12 M | 4.2 G | 2757 | 78.2 |
| SparX-Mamba (Lou et al., 2025) | 27 M | 5.2 G | 1370 | 83.5 |
| EfficientVMamba (Pei et al., 2025) | 33 M | 4.0 G | - | 81.8 |
| PlainMamba (Yang et al., 2024) | 25 M | 8.1 G | - | 81.6 |
| GroupMamba (Shaker et al., 2025) | 34 M | 7.0 G | 803 | 83.9 |
| VSSD (Shi et al., 2025) | 24 M | 4.5 G | - | 83.7 |
| DefMamba (Liu et al., 2025a) | 26 M | 4.8 G | - | 83.5 |
| MaIR (Li et al., 2025) | 26 M | 5.4 G | - | 83.1 |
| PRIS-Mamba (w/o PCF) | 27 M | 4.6 G | 2177 | 84.1 |
| PRIS-Mamba (ours) | 22 M | **3.9 G** | **2854** | **84.5** |

Introducing **partial channel filtering** yields further gains. PRIS-Mamba preserves only the informative channels for ring-based sequence processing and routes the remainder through a lightweight residual branch. This targeted allocation cuts computation from 4.6G to **3.9G** FLOPs, boosts throughput from 2177 to **2854** img/s (+30%), and also improves Top-1 from 84.1% to **84.5%** (+0.4%). The improvement in accuracy alongside reduced FLOPs indicates that the channel filter does more than prune cost: by steering the recurrent ring pathway toward high-signal channels, it produces cleaner ring descriptors and more effective radial composition. Compared to prior Mamba variants with similar or larger budgets (e.g, GroupMamba: 83.9% at 7.0G FLOPs; QuadMamba: 81.4% at 5.5G FLOPs), PRIS-Mamba attains higher accuracy with substantially lower compute and markedly higher realized throughput on A100.

In summary, the ablation isolates the contribution of channel filtering: selecting salient channels for the ring-by-ring traversal simultaneously reduces FLOPs, increases hardware throughput, and improves recognition accuracy. This supports our design choice to treat scan order and channel allocation as coupled levers for efficiency–accuracy optimization in Vision SSMs. Different scales models are provided in appendix.

### 4.2. Rotation Robustness of Ring vs. Fixed-Path Scans

Table 2 evaluates Top-1 accuracy under controlled in-plane rotations ($0°/30°/60°$). Fixed-path Vision-Mamba baselines uniformly degrade by $\approx 1$ to 2 points once the image is rotated, e.g, VMamba drops from 82.6 to 80.6 at $60°$, GroupMamba from 83.9 to 82.0, and PlainMamba from 81.6 to 79.8. In contrast, our ring-based scans are essentially flat across angles. PRIS-Mamba (w/o PCF) attains 84.1/83.9/83.9 at $0°/30°/60°$, indicating near-perfect rotation stability. Adding **partial channel filtering** further improves both the non-rotated baseline and the rotated cases:

*Table 2.* **Rotation stress test (Top-1, %).** We compare fixed-path Vision Mamba variants against our Ring Scan under no rotation and rotations of $30°/60°$ rendered on the same canvas.

| Model | No Rot. | 30° Rot. | 60° Rot. |
|---|---|---|---|
| Vim (Zhu et al., 2024a) | 80.5 | 78.6 (-1.9) | 78.6 (-1.9) |
| VMamba (Liu et al., 2024) | 82.6 | 80.7 (-1.9) | 80.6 (-2.0) |
| SiMBA (Patro & Agneeswaran, 2024) | 84.0 | 83.1 (-0.9) | 82.9 (-1.1) |
| Zigma (Hu et al., 2024) | 82.4 | 81.1 (-1.3) | 80.9 (-1.5) |
| QuadMamba (Xie et al., 2024) | 81.4 | 79.6 (-1.8) | 79.9 (-1.5) |
| LocalMamba (Huang et al., 2024) | 82.7 | 80.9 (-1.8) | 80.9 (-1.8) |
| FractalMamba (Xiao et al., 2025) | 83.0 | 81.9 (-1.1) | 81.7 (-1.3) |
| Adventurer (Wang et al., 2025b) | 78.2 | 76.5 (-1.7) | 76.8 (-1.4) |
| SparX-Mamba (Lou et al., 2025) | 83.5 | 82.1 (-1.4) | 82.3 (-1.2) |
| EfficientVMamba (Pei et al., 2025) | 81.8 | 79.7 (-2.1) | 79.9 (-1.9) |
| PlainMamba (Yang et al., 2024) | 81.6 | 79.6 (-2.0) | 79.8 (-1.8) |
| GroupMamba (Shaker et al., 2025) | 83.9 | 82.1 (-1.8) | 82.0 (-1.9) |
| VSSD (Shi et al., 2025) | 83.7 | 82.5 (-1.2) | 82.6 (-1.1) |
| DefMamba (Liu et al., 2025a) | 83.5 | 81.8 (-1.7) | 81.6 (-1.9) |
| MaIR (Li et al., 2025) | 83.1 | 81.5 (-1.6) | 81.4 (-1.7) |
| PRIS-Mamba (w/o PCF) | 84.1 | 83.9 | 83.9 |
| PRIS-Mamba (ours) | **84.5** | **84.3** | **84.4** |

PRIS-Mamba reaches **84.5** at $0°$ and remains at **84.3/84.4** under $30°/60°$, surpassing all prior variants at every angle.

Two key observations emerge. (1) The ring-by-ring alternating traversal (odd rings clockwise, even rings counterclockwise) preserves ring membership under rotation and converts global orientation changes into cyclic shifts along each closed ring, which do not harm the loop-aggregated descriptor. This explains the stability of PRIS-Mamba (w/o PCF) relative to fixed-path scans. (2) Partial channel filtering enhances ring descriptors by emphasizing high-signal channels for the recurrent ring path, improving overall accuracy without compromising rotation invariance. Together, these results support our central claim: *scan-order design*, combined with targeted channel allocation, resolves the sequence–geometry mismatch in path-based Vision SSMs, providing rotation robustness without specialized training.

### 4.3. COCO Detection and Instance Segmentation: Balancing Accuracy Efficiency

Table 3 reports Mask R-CNN results on MS COCO (minival, $1\times$ schedule, $1280\times800$) and shows that PRIS-Mamba advances both accuracy and efficiency relative to strong Vision-Mamba baselines. With only **235G** FLOPs, PRIS-Mamba achieves **48.9** AP$^{\text{box}}$ and **43.2** AP$^{\text{mask}}$, outperforming VMamba (46.5/42.1 at 262G) by $+2.4$ box AP and $+1.1$ mask AP while using $\sim$10% fewer FLOPs. Against Group-Mamba (47.6/42.9 at 279G) and DefMamba (47.5/42.8 at 268G), PRIS-Mamba delivers consistent gains $+1.3$ AP$^{\text{box}}$ and $+0.3\sim+0.4$ AP$^{\text{mask}}$ with lower costs.

Gains hold under stricter localization metrics (AP$^{\text{box}}_{75}$: **52.6** vs. 52.1/51.7), showing that the ring-by-ring alternating scan enhances precise box alignment rather than just recall (AP$_{50}$ also improves). Even against FLOP-heavier variants like PlainMamba-Adapter (542G), PRIS-Mamba achieves

*Table 3.* Mask R-CNN object detection and instance segmentation on MS COCO mini-val using 1× schedule. FLOPs are computed using input size $1280 \times 800$.

| Model | FLOPs (G) | Object Det. | | | Instance Seg. | | |
|---|---|---|---|---|---|---|---|
| | | AP$^{box}$ | AP$^{box}_{50}$ | AP$^{box}_{75}$ | AP$^{mask}$ | AP$^{mask}_{50}$ | AP$^{mask}_{75}$ |
| Vim | - | 45.7 | 63.9 | 49.6 | 39.2 | 60.9 | 41.7 |
| VMamba | 262 | 46.5 | 68.5 | 50.7 | 42.1 | 65.5 | 45.3 |
| QuadMamba | 301 | 46.7 | 69.0 | 51.3 | 42.4 | 65.9 | 45.6 |
| LocalMamba | 291 | 46.7 | 68.7 | 50.8 | 42.2 | 65.7 | 45.5 |
| FractalMamba | 266 | 46.8 | 68.7 | 50.8 | 42.4 | 65.9 | 45.8 |
| Adventurer | - | 46.5 | 65.2 | 50.4 | 40.3 | 62.2 | 43.5 |
| EfficientVMamba | - | 41.6 | 63.2 | 45.3 | 38.6 | 60.5 | 41.5 |
| PlainMamba-Adapter | 542 | 46.0 | 66.9 | 50.1 | 40.6 | 63.8 | 43.6 |
| GroupMamba | 279 | 47.6 | 69.8 | 52.1 | 42.9 | 66.5 | 46.3 |
| VSSD | 265 | 46.9 | 69.4 | 51.4 | 42.6 | 66.4 | 45.9 |
| DefMamba | 268 | 47.5 | 69.6 | 51.7 | 42.8 | 66.3 | 46.2 |
| PRIS-Mamba (ours) | **235** | **48.9** | **70.7** | **52.6** | **43.2** | **67.4** | **46.8** |

*Table 4.* Ablation of Partial Channel Filtering across Vision-Mamba families. Throughput (TP) values are measured with on Nvidia A100 GPU.

| Model | Params. | GFlops | TP (img/s) | Top-1 Acc% |
|---|---|---|---|---|
| Vim | 26 M | 5.3 G | 811 | 80.5 |
| +PCF | 23 M | 4.9 G (-0.4) | 1183 | 80.8 (+0.3%) |
| VMamba | 30 M | 5.6 G | 1686 | 82.6 |
| +PCF | 26 M | 5.1 (-0.5) G | 2314 | 82.9 (+0.3%) |
| QuadMamba | 31 M | 5.5 G | 1252 | 81.4 |
| +PCF | 25 M | 5.0 G (-0.5) | 1871 | 81.7 (+0.2%) |
| Adventurer | 12 M | 4.2 G | 2757 | 78.2 |
| +PCF | 10 M | 4.0 G (-0.2) | 3819 | 78.4 (+0.2%) |
| SparX-Mamba | 27 M | 5.2 G | 1370 | 83.5 |
| +PCF | 23 M | 4.8 G (-0.4) | 1959 | 83.9 (+0.4%) |
| GroupMamba | 34 M | 7.0 G | 803 | 83.9 |
| +PCF | 30 M | 6.4 G (-0.6) | 1075 | 84.2 (+0.3%) |
| PRIS-Mamba (w/o PCF) | 27 M | 4.6 G | 2177 | 84.1 |
| PRIS-Mamba (ours) | 22 M | 3.9 G (-0.7) | 2854 | 84.5 (+0.4%) |

higher accuracy with less than half the computational cost. These results show that careful scan-order design can deliver tangible improvements on downstream tasks without increasing model size or training schedule.

### 4.4. Generalization of Partial Channel Filtering

Table 4 examines whether mean-based *Partial Channel Filtering* (PCF) transfers beyond our architecture by inserting it into diverse Vision-Mamba backbones. Across Vim, VMamba, QuadMamba, Adventurer, SparX-Mamba, and GroupMamba, PCF consistently lowers computation and raises accuracy: parameters drop by 2–6M and FLOPs by 0.2–0.6G (roughly 4–12%), while throughput increases by about 30–45% and Top-1 improves by +0.2–0.4 pp. These gains are remarkably uniform across capacities and design variants, suggesting that (i) many channels contribute marginally to ring-style recurrent updates, and (ii) forwarding only high-activation channels through the recurrent path produces cleaner descriptors without starving the model of global context (residual branch).

Our models show the same pattern. PRIS-Mamba (w/o PCF) (84.1%, 4.6G, 2177 img/s); re-enabling PCF gives PRIS-Mamba (84.5%, 3.9G, 2854 img/s), *i.e*, $-15\%$ FLOPs,

*Table 5.* Comparing soft channel attention (SE, CBAM) with hard Partial Channel Filtering (PCF).

| Model | Params | GFlops | TP.(img/s) | Top-1 ACC% |
|---|---|---|---|---|
| PRIS-Mamba (w/o PCF) | 27 M | 4.6 G | 2177 | 84.1 |
| +SE | 27 M | 4.6 G (-) | 2089 | 84.2 (+0.1%) |
| PRIS-Mamba (w/o PCF) | 27 M | 4.6 G | 2177 | 84.1 |
| +CBAM | 30 M | 4.6 G (-) | 1982 | 84.3 (+0.2%) |
| PRIS-Mamba (w/o PCF) | 27 M | 4.6 G | 2177 | 84.1 |
| PRIS-Mamba (ours) | 22 M | 3.9 G (-0.7) | 2854 | 84.5 (+0.4%) |

$\sim$ +30% throughput, and +0.4 pp Top-1. Notably, accuracy increases despite reduced compute, indicating that PCF does more than prune cost: it improves the signal-to-noise ratio of ring descriptors and stabilizes the subsequent radial composition. Together, these results support PCF as a simple, general plug-in for Vision-SSMs that advances the accuracy–efficiency frontier without architectural surgery or re-training tricks.

### 4.5. Comparing soft channel attention (SE, CBAM) with hard Partial Channel Filtering (PCF)

Table 5 isolates the effect of channel selection strategy on the same backbone. Compared to PRIS-Mamba (w/o PCF) at 27M/4.6G/2177 img/s/84.1%, adding **SE** (Hu et al., 2018) preserves the computation but slows inference (2089 img/s) for a marginal +0.1 pp. **CBAM** (Woo et al., 2018) further increases parameters (30M) and slows throughput (1982 img/s) for +0.2 pp, again without FLOP reduction. By contrast, **PCF** (ours) delivers a strictly better accuracy–efficiency point: PRIS-Mamba reaches 22M params, 3.9G FLOPs, 2854 img/s, and 84.5% Top-1, simultaneously *reducing* compute/params and *increasing* accuracy/throughput. These results support the premise of Sec. 3.5: when the downstream ring module operates recurrently, forwarding only high-salience channels (and bypassing the rest) is more effective than soft reweighting that still processes all channels.

### 4.6. Random-Mask Occlusion: Local Damage vs. Traversal Robustness

This experiment evaluates robustness to *local missing content* by randomly zeroing a square tile while preserving the input size. Unlike rotations or patch shuffling, occlusion removes content without permuting neighborhoods. Fixed-path Vision Mamba variants exhibit consistent accuracy drops, up to 1.3 points with $16\times16$ masking (Table 6), with larger masks causing greater degradation due to long contiguous scan segments traversing missing tokens and propagating weakened recurrent states.

**PRIS-Mamba** is notably more resilient (drops $\leq 0.6$). Two factors contribute. (i) *Order-agnostic ring aggregation* dilutes the influence of a localized hole: a single masked tile affects only the rings it intersects and is averaged with

*Table 6.* **Random-mask occlusion stress test (Top-1, %).** We randomly drop a contiguous square region at test time ("Mask $4\times4$": one of $2\times2$ tiles is zeroed; "Mask $16\times16$": one of $4\times4$ tiles is zeroed). Numbers in parentheses are absolute drops from the unmasked Top-1. Ring-Mamba denotes our Ring Scan module plugged into a Vision-Mamba backbone.

| Model | Params | Top-1 | Mask $4\times4$ | Mask $16\times16$ |
|---|---|---|---|---|
| VMamba | 30M | 82.6% | 82.3% (-0.3) | 81.7% (-0.9) |
| | 50M | 83.6% | 83.2% (-0.4) | 82.3% (-1.3) |
| LocalMamba-T | 30M | 82.7% | 82.3% (-0.4) | 81.9% (-0.8) |
| | 50M | 83.7% | 83.3% (-0.4) | 82.5% (-1.2) |
| **PRIS-Mamba (ours)** | 30M | **84.5%** (-0.1) | **84.4%** (-0.1) | **84.1%** (-0.4) |
| | 50M | **85.3%** | **85.2%** (-0.1) | **84.7%** (-0.6) |

many valid pixels in those rings. (ii) *Radial state-space integration* transports information across rings rather than along a fragile global path; mask-gated updates further attenuate rings dominated by the occluded region, preventing corrupted states from spreading. Together, the ring-wise summarization and radial propagation maintain stable performance under local erasures, complementing the rotation and patch-order robustness shown in the other stress tests.

## 5. Conclusion

We revisited the often-overlooked role of scan order in Vision State Space Models and proposed **PRIS-Mamba**, a rotation-robust traversal that aggregates features ring by ring and propagates context radially, enhanced with lightweight *partial channel filtering*. This design maintains SSMs' linear-time efficiency while better aligning sequential token adjacency with 2D geometry, addressing a fundamental source of performance degradation in path-based Vision SSMs. Empirically, PRIS-Mamba sets a new accuracy–efficiency frontier: on ImageNet-1K, it reaches 84.5% Top-1 with 3.9G FLOPs and 2854 img/s on A100, surpassing VMamba-T. On COCO, it attains 48.9 AP[box] and 43.2 AP[mask] at 235G FLOPs, outperforming strong Vision-Mamba baselines with less computation. Under rotation stress, our ring traversal remains stable while fixed-path scans drop by about 1–2%.

**Limitations:** Our current implementation relies on a fixed image center and a discrete ring width, which may be suboptimal for off-center subjects or images with extreme aspect ratios. While the method can become object-aware if an object detector provides the center, severe rotations that create large padded regions still reduce valid information, potentially impacting performance.

**Future work** includes learning ring origins and widths, as well as designing content-adaptive ring partitions. Coupling ring traversal with learned inpainting priors or anti-aliasing warps may further improve robustness under extreme rotations. Extensions to spatio–temporal rings for video, 3D medical volumes, and object-aware traversals

are also promising directions. We hope this work encourages further exploration of traversal design as a low-cost, principled approach to robust and efficient Vision SSMs.

## Acknowledgements

Kuan-Chuan Peng was exclusively supported by Mitsubishi Electric Research Laboratories. The other authors sincerely thank the National Science and Technology Council (NSTC) of Taiwan under Grant No. NSTC 115-2634-F-A49-014 -.

## Impact Statement

This paper presents work whose primary goal is to advance the field of Machine Learning, particularly efficient and robust visual representation learning with state space models. By improving the scan-order design of Vision SSMs, our method may contribute to more reliable and computationally efficient visual recognition systems, which can be beneficial for applications requiring robustness to geometric variations and limited computational resources. At the same time, as with many general-purpose vision backbones, the proposed model could be incorporated into downstream systems such as surveillance, remote sensing, or automated decision-making pipelines. Such deployments should be accompanied by appropriate data governance, privacy protection, bias evaluation, and human oversight. We do not identify additional ethical risks that are specific to the proposed scan mechanism beyond the broader societal considerations associated with advancing visual recognition technologies.

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

# Partial Ring Scan: Revisiting Scan Order in Vision State Space Models
## Supplementary Material

In this supplementary material, we provide additional experiments and analyses to further illustrate the design choices and robustness of **PRIS-Mamba**. The contents are organized as follows:

- § A details the PRIS-Mamba algorithm, including Partial Channel Filtering (PCF), ring-by-ring alternating scan, radial aggregation, and the final $1\times1$ projection/fusion.

- § B presents ImageNet-1K classification results, emphasizing accuracy–efficiency trade-offs across model scales and comparisons to Vision-Mamba baselines.

- § C studies *scan-order ablations* on a fixed PRIS-Mamba backbone (S1–S21 in Fig. 6), quantifying how different traversals affect throughput, accuracy, and rotation robustness.

- § D reports an ablation on ring width $\Delta r$, showing stable performance over a wide range and identifying a default that balances detail preservation and computational cost.

- § E evaluates *patch-order shuffling* (Fig. 7), demonstrating robustness to token reordering due to order-agnostic ring aggregation and short radial SSMs.

- § F analyzes XY-plane rotation sensitivity on a preserved canvas, highlighting the geometric limits of fixed-path scans and the rotation stability of ring-wise traversal.

- § G examines *object-aware coupling* (via YOLOv12) versus centered ring scan, clarifying the speed–accuracy trade-off and the complementary role of PCF.

These sections together provide a comprehensive view of PRIS-Mamba's design choices, efficiency characteristics, and robustness under challenging perturbations, complementing the main paper with actionable ablations and guidelines.

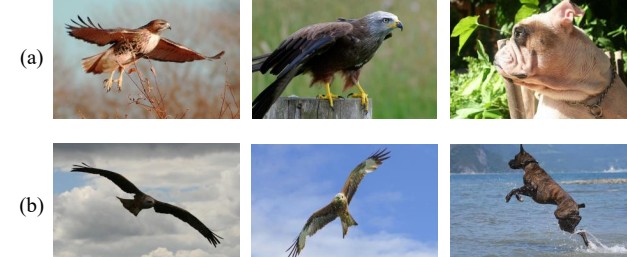

*Figure 5.* **Upright vs. in-the-wild rotations.** (a) Canonical, upright views typical of standard training; (b) naturally rotated or tilted instances that occur in practice.

## Why PRIS-Mamba is useful?

PRIS-Mamba is useful because it tackles two long-standing, under addressed issues at once: expanding spatial receptive context and maintaining robustness to rotation. Prior Vision-Mamba models typically widen context by stacking multiple fixed scan paths; as Tab. 8 shows, adding paths reliably improves accuracy but steadily erodes throughput, creating a hard efficiency ceiling. In real scenario, rotations and tilts are common; fixed-path scans then break token continuity and degrade accuracy. PRIS-Mamba groups pixels by radius into concentric rings and applies order agnostic aggregation within each ring, with short radial SSMs propagating context across rings. This preserves the token count and computational profile of conventional V-SSMs, avoiding the efficiency penalties of multi-path designs while remaining inherently stable under in-plane rotations (ring membership is unchanged; only the starting angle shifts). Coupled with Partial Channel Filtering, which routes only high-salience channels through the recurrent ring pathway, PRIS-Mamba not only reduces compute and memory but increases both throughput and accuracy. In short, *PRIS-Mamba* achieves multi-path-level spatial coverage with single-path efficiency and rotation robustness, making it better suited for real-world use.

## A. The PRIS-Mamba Algorithm

Algorithm 1 summarizes our pipeline. Given a feature map $X \in \mathbb{R}^{H \times W \times C}$, PRIS-Mamba first applies *Partial Channel Filtering* (PCF) to route informative channels, then performs

**Algorithm 1** PRIS-Mamba with Partial Channel Filtering

1: **Input:** $X \in \mathbb{R}^{H \times W \times C}$;
2: **Partial Channel Filtering (PCF)**
3: $\mu_i \leftarrow \mathrm{GAP}(X[:,:,i])$ for $i=1..C$; $\mu \leftarrow \frac{1}{C}\sum_{i=1}^{C}|\mu_i|$;
   // GAP=Global Average Pooling;
4: $\mathbf{M}_{\mathrm{part}} \leftarrow \{ i : \mu_i \geq \mu \}$; $\quad \mathbf{M}_{\mathrm{id}} \leftarrow \{ i : \mu_i < \mu \}$;
5: $X_{\mathrm{part}} \leftarrow X[:,:,\mathbf{M}_{\mathrm{part}}]$; $\quad X_{\mathrm{id}} \leftarrow X[:,:,\mathbf{M}_{\mathrm{id}}]$;
6: **Ring Indexing**
7: $(c_x, c_y) \leftarrow$ image center;
8: $\hat{r}(u,v) \leftarrow \lfloor \|(u-c_x, \, v-c_y)\|_2 \, / \, \Delta r \rfloor$ for all $(u,v)$;
9: $R^\star \leftarrow \max_{u,v} \hat{r}(u,v)$;
10: **Ring-by-Ring Scan**
    clockwise (CW) / counterclockwise (CCW);
11: **for** $r=0..R^\star$ **do**
12: $\quad \Omega_r \leftarrow \{(u,v) : \hat{r}(u,v)=r\}$;
12: $\quad$ **if** $r$ is even **then**
13: $\quad\quad \sigma_r \leftarrow \mathrm{ContourOrder}(\Omega_r, \, \mathrm{dir}=\mathrm{CW})$;
13: $\quad$ **else**
14: $\quad\quad \sigma_r \leftarrow \mathrm{ContourOrder}(\Omega_r, \, \mathrm{dir}=\mathrm{CCW})$;
14: $\quad$ **end if**
15: $\quad x_{r,k} \leftarrow \mathrm{Conv}_{1\times1}\big(X_{\mathrm{part}}[\sigma_r(k)]\big)$ for $k=1..|\Omega_r|$;
16: $\quad z_r \leftarrow \mathrm{SSM\_AlongLoop}\big(\{x_{r,k}\}_{k=1}^{|\Omega_r|}\big)$;
17: **end for**
18: **Radial SSM (Inner → Outer)**
19: $\{y_r^{\mathrm{rad}}\}_{r=0}^{R^\star} \leftarrow \mathrm{SSM\_Radial}\big(\{z_r\}_{r=0}^{R^\star}\big)$;
20: **Write-Back & Fusion**
21: $Y_{\mathrm{part}} \leftarrow \mathrm{Conv}_{1\times1}\big(y_{\hat{r}(u,v)}^{\mathrm{rad}}\big)$;
22: $\hat{Y} \leftarrow X_{\mathrm{part}} + \mathrm{Conv}_{1\times1}(Y_{\mathrm{part}})$;
23: $Y \leftarrow \mathrm{ConcatChannels}(\hat{Y}; \mathbf{M}_{\mathrm{id}})$;
24: **Return:** output $Y \in \mathbb{R}^{H \times W \times C}$. =0

*Table 7.* Performance comparison on ImageNet-1K with $224 \times 224$ inputs. Throughput (TP) is measured on an Nvidia A100 GPU, and Top-1 Accuracy is reported in %. Suffixes **T/S/B** denote Tiny, Small, and Big variants.

| Model | Params. | GFlops | TP (img/s) | Top-1 Acc. |
|---|---|---|---|---|
| Vim-S (Zhu et al., 2024a) | 26 M | 5.3 G | 811 | 80.5 |
| VMamba-T (Liu et al., 2024) | 30 M | 5.6 G | 1686 | 82.6 |
| SiMBA-S (Patro & Agneeswaran, 2024) | 27 M | 5.0 G | - | 84.0 |
| Zigma-S (Hu et al., 2024) | 31 M | 5.1 G | - | 82.4 |
| QuadMamba-S (Xie et al., 2024) | 31 M | 5.5 G | 1,252 | 81.4 |
| LocalVMamba-T (Huang et al., 2024) | 26 M | 5.7 G | - | 82.7 |
| FractalMamba-T (Xiao et al., 2025) | 31 M | 4.8 G | - | 83.0 |
| Adventurer-T (Wang et al., 2025b) | 12 M | 4.2 G | 2,757 | 78.2 |
| SparX-Mamba-T (Lou et al., 2025) | 27 M | 5.2 G | 1,370 | 83.5 |
| EfficientVMamba-B (Pei et al., 2025) | 33 M | 4.0 G | - | 81.8 |
| PlainMamba-T (Yang et al., 2024) | 25 M | 8.1 G | - | 81.6 |
| GroupMamba-S (Shaker et al., 2025) | 34 M | 7.0 G | 803 | 83.9 |
| VSSD-T (Shi et al., 2025) | 24 M | 4.5 G | - | 83.7 |
| DefMamba-S (Liu et al., 2025a) | 26 M | 4.8 G | - | 83.5 |
| MaIR-T (Li et al., 2025) | 26 M | 5.4 G | - | 83.1 |
| PRIS-Mamba-T (w/o PCF) | 27 M | 4.6 G | 2,177 | 84.1 |
| PRIS-Mamba-T | 22 M | **3.9 G** | **2,854** | **84.5** |
| VMamba-S (Liu et al., 2024) | 50 M | 11.2 G | 877 | 83.6 |
| SiMBA-B (Patro & Agneeswaran, 2024) | 40 M | 9.0 G | - | 84.7 |
| QuadMamba-B (Xie et al., 2024) | 50 M | 9.3 G | 582 | 83.8 |
| LocalVMamba-S (Huang et al., 2024) | 50 M | 11.4 G | - | 83.7 |
| FractalMamba-S (Xiao et al., 2025) | 54 M | 9.6 G | - | 83.6 |
| Adventurer-S (Wang et al., 2025b) | 44 M | 8.3 G | 1405 | 81.8 |
| SparX-Mamba-S (Lou et al., 2025) | 47 M | 9.3 G | - | 84.2 |
| PlainMamba-S (Yang et al., 2024) | 50 M | 14.4 G | - | 82.3 |
| GroupMamba-B (Shaker et al., 2025) | 57 M | 14.0 G | - | 84.5 |
| VSSD-S (Shi et al., 2025) | 40 M | 7.4 G | - | 84.1 |
| DefMamba-B (Liu et al., 2025a) | 50 M | 8.5 G | - | 84.2 |
| PRIS-Mamba-S (w/o PCF) | 50 M | 7.6 G | 1,165 | 84.4 |
| PRIS-Mamba-S | 44 M | **6.7 G** | **1,679** | **84.9** |

a *ring-by-ring alternating scan* with a selective SSM along each ring, followed by a *short radial SSM* that aggregates near-to-far context across rings, and finally projects the result back to the grid via a $1\times1$ projection.

## B. ImageNet-1K Results: Demonstrating Accuracy and Efficiency Gains

The goal of these experiments is to show that PRIS-Mamba delivers higher accuracy while reducing computation and increasing inference speed, compared with existing Vision-Mamba baselines and recent state-of-the-art models.

Table 7 reports single-crop Top-1 accuracy, FLOPs, and throughput on ImageNet-1K ($224\times224$) running on an A100 GPU. We evaluate two practical model sizes: "tiny" models referred in the main paper ($\sim$25–30M params) and small to large models ($\sim$45–50M). In the table, suffixes **T/S/B** correspond to Tiny, Small, and Big variants.

On the 'tiny' scale, **PRIS-Mamba-T** attains 84.5% Top-1 with 3.9 FLOPs and 2,854img/s, surpassing VMamba-T (82.6%, 5.6G, 1,686 img/s) by 1.5% while using $\approx$30% fewer FLOPs and roughly 40% faster inference. Removing Partial Channel Filtering (PCF), PRIS-Mamba-T (w/o PCF) already yields 84.1% at 4.6G and 2,177 img/s; adding PCF reduces computation by 0.7G and cuts parameters from 27M to 22M, while still increase 0.4% Top-1 accuracy and increase throughput to 2,854 img/s due to the effects of important channel selection and filtering.

At the 'small' scale, **PRIS-Mamba-S** reaches 84.9% with 6.7 FLOPs and 1,679 img/s, outperforming VMamba-S (83.6%, 11.2G, 877 img/s). This corrresponds to +1.3 Top-1 Acc%, over 40% fewer FLOPs and nearly double the throughput with fewer parameters: 44M vs. 50M. Compared with recent contemporaries such as SiMBA-B (84.7%, 9.0G) and GroupMamba-B (84.5%, 14.0G), PRIS-Mamba-S reaches state-of-the-art accuracy with the best efficiency among similarly sized models.

Overall, across both model sizes, PRIS-Mamba consistently improves the accuracy-efficiency balance, providing higher accuracy, lower FLOPs, and faster inference compared with both Vision-Mamba baselines and recent strong alternatives.

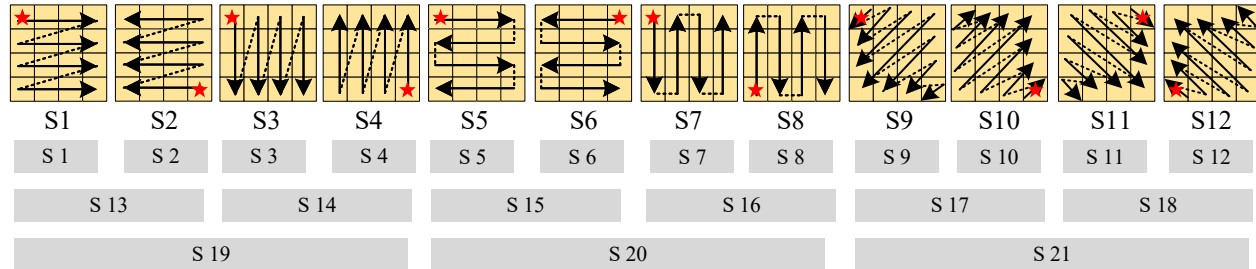

*Figure 6.* **Primitive scan orders.** Twelve canonical paths (S1–S12) such as left-to-right raster, serpentine, and diagonal produce distinct 1D sequences from the same image. S1-12 evaluate single scans; S13–18 evaluate pairs of scans; S 19–21 aggregate four scans, enabling a systematic comparison of scan-order effects.

## C. Scan-Order Ablations on a Fixed PRIS-Mamba Backbone

We evaluate *the same PRIS-Mamba architecture* while varying only its scan order (S1–S21 in Fig. 6). As reported in Tab. 8, single-path scans (S1–S12) deliver 81.9–83.1% Top-1 at ~1.86k img/s. Pairing two paths (S13–S18) improves accuracy to 82.6–83.5% but reduces throughput to ~1.70k img/s; aggregating four paths (S19–S21) further raises accuracy up to 84.1% with another drop to ~1.60k img/s. In contrast, **PRIS-Mamba** with our ring-by-ring alternating scan and PCF achieves **84.9%** Top-1 at **1,679 img/s**, surpassing the best four-path baseline in accuracy while retaining higher throughput. The rotation stress test in Tab. 9 shows that all fixed-path variants (S1–S21) incur consistent losses of ~1.2–1.9 percentage points at $30°/60°$, indicating that stacking more paths does not resolve the sequence–geometry mismatch. By contrast, PRIS-Mamba remains essentially unchanged (84.9% → 84.8/84.9%), demonstrating that *scan-order design*—rather than path count—governs both the accuracy–throughput frontier and robustness to rotation.

## D. Ablation on Ring Width

We analyze the effect of the ring width $\Delta r$ (in pixels) that controls how finely the image is divided into concentric rings. Reducing $\Delta r$ increases the number of rings $R \approx \lfloor r_{\max}/\Delta r \rfloor + 1$, thereby lengthening the radial sequence $T = R + 1$ for the state-space update. Conversely, increasing $\Delta r$ merges more pixels per ring and reduces spatial granularity.

Table 10 shows that PRIS-Mamba remains stable across a wide range: $\Delta r = 1, 2, 4$ differ by at most $0.2\%$ Top-1 points. The mild peak at $\Delta r = 2$ reflects a balance between: (i) preserving adequate per-ring local detail (very large $\Delta r$) blurs near-center information and (ii) keeping the radial sequence compact and well-conditioned (very small $\Delta r$ increases $R$ and introduces noisier statistics), where $R$ is the number of rings. Since the radial pass scales as $O(R)$, $\Delta r = 2$ yields the best accuracy–efficiency trade-off and is

*Table 8.* Performance comparison among different scanning methods on ImageNet-1K with $224 \times 224$ inputs. Throughput (TP) is measured on Nvidia A100 GPU, and Top-1 Accuracy is reported in %.

| Model | GFlops | TP (img/s) | Top-1 Acc. |
|---|---|---|---|
| S 1 | 6.7 G | 1863 | 82.1 |
| S 2 | 6.7 G | 1854 | 81.9 |
| S 3 | 6.7 G | 1867 | 82.0 |
| S 4 | 6.7 G | 1870 | 82.1 |
| S 5 | 6.7 G | 1878 | 82.4 |
| S 6 | 6.7 G | 1865 | 82.4 |
| S 7 | 6.7 G | 1870 | 82.3 |
| S 8 | 6.7 G | 1858 | 82.6 |
| S 9 | 6.7 G | 1823 | 83.0 |
| S 10 | 6.7 G | 1827 | 82.8 |
| S 11 | 6.7 G | 1816 | 83.1 |
| S 12 | 6.7 G | 1823 | 82.9 |
| S 13 | 6.7 G | 1718 | 82.6 |
| S 14 | 6.7 G | 1709 | 82.8 |
| S 15 (Zigma (Hu et al., 2024)) | 6.7 G | 1712 | 83.1 |
| S 16 | 6.7 G | 1707 | 83.0 |
| S 17 | 6.7 G | 1677 | 83.5 |
| S 18 | 6.7 G | 1668 | 83.3 |
| S 19 (Vmamba (Liu et al., 2024)) | 6.7 G | 1622 | 83.3 |
| S 20 (PlainMamba (Yang et al., 2024)) | 6.7 G | 1626 | 83.5 |
| S 21 | 6.7 G | 1597 | 84.1 |
| PRIS-Mamba-S (w/o PCF) | 7.6 G | 1,165 | 84.4 |
| PRIS-Mamba-S | 6.7 G | 1,679 | **84.9** |

adopted as the default.

## E. Patch-Order Shuffling: Evaluating Robustness to Token Reordering

We investigate how strongly fixed-path Vision Mamba models depend on token order using a *patch-order shuffling* stress test. The input image is divided into tiles that are randomly permuted before inference, with difficulty levels: Fig. 7: (a) the original input, (b) the image split into 2×2 non-overlapping tiles ("cut-4") and randomly permuted, and (c) the image split into 4×4 tiles ("cut-16") and randomly permuted. This preserves local visual content within each tile while deliberately breaking long contiguous scan seg-

*Table 9.* **Rotation stress tests and comparisons under various scanning methods.** We compare different scan variants against our Ring Scan under no rotation and rotations of $30°/60°$ rendered on the same canvas.

| Model | No Rot. | 30° Rot. | 60° Rot. |
|---|---|---|---|
| S 1 | 82.1 | 80.6 (-1.5) | 80.4 (-1.7) |
| S 2 | 81.9 | 80.3 (-1.6) | 80.4 (-1.5) |
| S 3 | 82.0 | 80.3 (-1.7) | 80.5 (-1.5) |
| S 4 | 82.1 | 80.5 (-1.6) | 80.6 (-1.5) |
| S 5 | 82.4 | 80.8 (-1.6) | 80.9 (-1.5) |
| S 6 | 82.4 | 80.7 (-1.7) | 80.6 (-1.8) |
| S 7 | 82.3 | 80.8 (-1.5) | 80.5 (-1.8) |
| S 8 | 82.6 | 80.9 (-1.7) | 80.7 (-1.9) |
| S 9 | 83.0 | 81.5 (-1.5) | 81.3 (-1.7) |
| S 10 | 82.8 | 81.4 (-1.4) | 81.5 (-1.3) |
| S 11 | 83.1 | 81.4 (-1.7) | 81.6 (-1.5) |
| S 12 | 82.9 | 81.5 (-1.4) | 81.3 (-1.6) |
| S 13 | 82.6 | 81.3 (-1.3) | 81.2 (-1.4) |
| S 14 | 82.8 | 81.5 (-1.3) | 81.3 (-1.5) |
| S 15 (Zigma (Hu et al., 2024)) | 83.1 | 81.6 (-1.5) | 81.5 (-1.6) |
| S 16 | 83.0 | 81.4 (-1.6) | 81.7 (-1.3) |
| S 17 | 83.5 | 81.9 (-1.6) | 82.0 (-1.5) |
| S 18 | 83.3 | 81.8 (-1.5) | 81.9 (-1.4) |
| S 19 (Vmamba (Liu et al., 2024)) | 83.3 | 82.1 (-1.2) | 82.2 (-1.1) |
| S 20 (PlainMamba (Yang et al., 2024)) | 83.5 | 82.4 (-1.1) | 82.5 (-1.0) |
| S 21 | 84.1 | 82.8 (-1.3) | 82.7 (-1.4) |
| PRIS-Mamba | **84.9** | **84.8** | **84.9** |

*Table 10.* **Ablation on ring width $\Delta r$ w.r.t Top-1 Accuracy in %.** We vary the ring width $\Delta r \in \{1, 2, 4\}$ pixels in our PRIS-Mamba while keeping all other settings fixed.

| Model | Params | $\Delta r = 1$ | $\Delta r = 2$ | $\Delta r = 4$ |
|---|---|---|---|---|
| PRIS-Mamba-T | 30M | 84.3% (-0.2) | 84.5% | 84.3% (-0.2) |
| PRIS-Mamba-S | 50M | 84.8% (-0.1) | 84.9% | 84.7% (-0.2) |

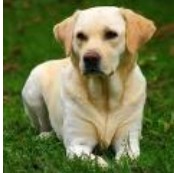 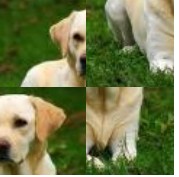 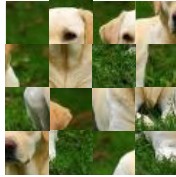

(a) Original    (b) cut 4 patches    (c) cut 16 patches

*Figure 7.* **Patch-order shuffling ablation.** We study scan-order sensitivity by permuting the order of non-overlapping patches. (a) Original image; (b) image divided into $2\times2$ patches and randomly shuffled; (c) image divided into $4\times4$ patches and randomly shuffled. Performance changes isolate the effect of inter-patch adjacency on sequence models.

*Table 11.* **Patch-order shuffling stress test:** We evaluate robustness of PRIS-Mamba when the image is tiled and the patches are randomly permuted: "Cut 4 patches" has $2\times2$ tiles; "Cut 16 patches" has $4\times4$ tiles. Baselines are fixed-path Vision Mamba models, while our method uses Ring Scan with a *dynamic random step* during training to prevent overfitting to one fixed radial layout. Numbers in parentheses show drops relative to the non-shuffled Top-1.

| Model | Params | Top-1 | Cut 4 patches | Cut 16 patches |
|---|---|---|---|---|
| VMamba (Baseline) | 30M | 82.6% | 82.1% (-0.5) | 81.3% (-1.3) |
|  | 50M | 83.6% | 82.9% (-0.7) | 81.9% (-1.8) |
| LocalVMamba-T | 30M | 82.7% | 82.2% (-0.5) | 81.5% (-1.2) |
|  | 50M | 83.7% | 82.9% (-0.8) | 82.1% (-1.6) |
| **PRIS-Mamba** | 30M | **84.5%** | **84.3%** (-0.2) | **83.8%** (-0.7) |
|  | 50M | **85.3%** | **84.9%** (-0.4) | **84.3%** (-1.0) |

ments and inter-tile adjacency learned by ordered traversals.

Table 11 shows that ordered-scan baselines (VMamba, LocalVMamba-T) experience consistent accuracy drops of 0.5-1.8 points as the shuffle becomes more severe. This reflects a structural limitation: masking can suppress invalid or inconsistent tokens but cannot restore correct adjacency once tiles are permuted relative to the learned scan direction. In contrast, our PRIS-Mamba maintains strong accuracy with much smaller drops ($\leq 1.0$ point). This robustness arises from two architectural choices. First, ring-wise aggregation is order-agnostic within each ring, so reordering tiles does not corrupt the per-ring representation. Second, the radial update proceeds along the ring index rather than a fragile global traversal, avoiding dependence on a single topological scan path. We also apply a *dynamic random step* during training, randomly varying ring width and stride within a small range each iteration. This prevents the model from overfitting to one fixed discretization and further improves stability under shuffled and rotated inputs.

Overall, the results show that the traversal strategy itself is the key factor in achieving robustness to token-order perturbation: fixed-path Mamba variants are intrinsically sensitive to reordered inputs, while ring-wise aggregation combined with radial state-space modeling largely eliminates this failure mode.

## F. XY-Plane Rotation Impact on ImageNet-1K Classification

Table 12 evaluates how well scan-based Vision Mamba backbones handle in-plane rotations while keeping the output frame unchanged. For small rotations ($15°$), all models behave similarly because scan continuity is only slightly disrupted and padding is minimal. At around $60°$, fixed-path variants (VMamba, LocalVMamba-T) begin to show noticeable drops (about 0.8–1.5 points), indicating that their assumed token adjacency no longer matches the underlying spatial structure. In contrast, our PRIS-Mamba remains stable at this level (drops $\leq 1.1$) because ring membership is unaffected by global orientation and within-ring aggregation does not depend on angular order.

At rotation of $75°$, performance decreases sharply for all models, including our PRIS-Mamba. This is driven by two main effects: (i) large rotations introduce extensive zero padding, reducing the effective field of view and available evidence; and (ii) fixed-path scans additionally suffer from

*Table 12.* **Evaluation of XY-plane rotation on impact on ImageNet-1K (Top-1 Acc. %):** We evaluate fixed-path Vision Mamba variants and PRIS-Mamba under in-plane rotations of $15°$, $60°$, and $75°$ (canvas preserved; padded corners zeroed). Parentheses indicate absolute drops relative to the non-rotated baseline. Two capacities are reported per model (second row shows the larger). Small rotations ($15°$) have minimal effect; around $60°$, fixed-path scans begin to degrade; at $75°$, all models drop sharply, but PRIS-Mamba remains the most accurate.

| Model | 0° | 15° | 60° | 75° |
|---|---|---|---|---|
| VMamba | 82.6% | 82.5% (-0.1) | 81.8% (-0.8) | 68.2% (-14.4) |
| | 83.6% | 83.6% (0.0) | 82.2% (-1.4) | 70.6% (-13.0) |
| LocalVMamba-T | 82.7% | 82.6% (-0.1) | 81.8% (-0.9) | 67.8% (-14.9) |
| | 83.7% | 83.5% (-0.2) | 82.2% (-1.5) | 70.4% (-13.3) |
| **PRIS-Mamba** | 84.8% | 84.7% (-0.1) | 84.2%(-0.6) | 70.7% (-14.1) |
| | **85.8%** | **85.8%** (0.0) | **84.7%** (-1.1) | **73.2%** (-12.6) |

*Table 13.* Object-aware coupling (YOLOv12) and PCF on PRIS-Mamba-S. Throughput (TP) is measured on Nvidia A100 GPU, and Top-1 Accuracy is reported in %.

| Model | Yolov12 | GFlops | TP (img/s) | Top-1 Acc. |
|---|---|---|---|---|
| VMamba-S (Liu et al., 2024) | ✗ | 11.2 G | 877 | 83.6 |
| PRIS-Mamba-S (w/o PCF) | ✗ | 7.6 G | 1,165 | 84.4 |
| PRIS-Mamba-S (w/o PCF) | ✓ | 8.9 G | 678 | 84.7 |
| PRIS-Mamba-S | ✗ | **6.7 G** | **1,679** | 84.9 |
| PRIS-Mamba-S | ✓ | 7.8 G | 1,027 | **85.3** |

severe adjacency mismatch, compounding errors propagated along the scan direction. PRIS-Mamba still performs best but drops by 12.6 to 14.1 points, consistent with loss of visual information rather than a breakdown of the traversal design.

Overall, the results suggest a practical geometric limit near $60°$ for fixed-path models, while ring-wise traversal extends this threshold by preserving spatial structure under rotation.

## G. Object-Aware vs. Centered Ring Scan

Table 13 compares VMamba-S and our PRIS-Mamba-S under two traversal settings: centered ring scan and an object-aware variant coupled with YOLOv12, and with or without Partial Channel Filtering (PCF). Under matched training and input resolution, PRIS-Mamba-S (w/o PCF) attains 84.4% Top-1 at 7.6G FLOPs with 1165 img/s, whereas VMamba-S reports 83.6% at 11.2G and 877 img/s; enabling PCF further improves PRIS-Mamba-S to 84.9% at 6.7G with 1679 img/s, indicating higher accuracy and throughput alongside reduced compute. Introducing object-aware coupling increases accuracy but raises cost: without PCF, Top-1 moves from 84.4% to 84.7% while FLOPs rise (7.6G to 8.9G) and throughput decreases (1165 to 678 img/s); with PCF, the object-aware variant achieves the highest Top-1 of 85.3%, though it remains heavier and slower (7.8G, 1027 img/s) than the centered-with-PCF configuration (6.7G, 1679 img/s, 84.9%). Across both settings, PCF consistently reduces FLOPs and increases throughput while yielding modest accuracy gains.

These results suggest a practical guideline. For *throughput-critical* deployments (*e.g*, real-time or edge settings), the centered ring scan with PCF offers the best speed/compute profile while preserving strong accuracy. When *peak accuracy* is the priority and additional compute is acceptable, the object-aware ring scan is preferred, as it aligns traversal with object geometry at the expense of FLOPs and throughput.

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
