# OpenReview forum: "Partial Ring Scan: Revisiting Scan Order in Vision State Space Models"
_ICML.cc/2026/Conference — ICML 2026 regular_

### Official Review · Reviewer_Fs6E · 2026-02-28

**Soundness:** 4
**Presentation:** 4
**Significance:** 2
**Originality:** 2
**Overall Recommendation:** 4
**Confidence:** 2

**Summary:**

This paper investigates the scan order used in Vision State Space Models (SSMs). The authors propose Partial Ring Scan Mamba (PRIS-Mamba), which decomposes images into concentric rings and aggregates features within each ring. This design preserves spatial isotropy and achieves rotation robustness without requiring polar remapping or rotation-specific training, while maintaining the linear-time complexity of SSMs. To further improve efficiency, the method introduces Partial Channel Filtering, which routes only informative channels through recurrent ring-based processing and sends the remaining channels through a lightweight residual path. Extensive experiments on ImageNet-1K and MS COCO demonstrate that PRIS-Mamba outperforms prior Vision SSMs in terms of accuracy, throughput, and FLOPs. Notably, the model maintains nearly constant performance under in-plane rotations, whereas fixed-path scan methods degrade significantly.

**Compliance With Llm Reviewing Policy:**

Affirmed.

**Final Justification:**

The proposed ring scanning pattern and partial channel filtering appear to be motivated primarily by empirical observations. This raises concerns about their generalizability to higher-dimensional inputs (e.g., 3D or 4D sequences, such as those in humanoid robotics), and whether these design choices are sufficiently justified to merit acceptance.

Overall, the paper demonstrates strong empirical results but lacks solid theoretical grounding for the proposed SSM scanning pattern. I will increase my score, but lower my confidence in this assessment, leaving the final judgment to the ACs and PCs.

**Key Questions For Authors:**

- How does the proposed Ring Scan interact with the underlying SSM dynamics?
Can the authors provide a formal or intuitive analysis that connects ring-wise aggregation and radial SSM composition to the SSM state update equations, beyond empirical observations?
- What properties of Ring Scan make it more compatible with SSM recurrence than prior scan orders?
For example, does Ring Scan improve state stability, effective receptive field growth, or signal mixing compared to raster or serpentine scans?
- Can the authors provide a detailed analysis of the computational complexity and memory access patterns of Ring Scan?
In particular, does the throughput improvement observed in Table 1 (PRIS-Mamba without PCF vs. VMamba) primarily stem from the Ring Scan design itself?
- Can the authors include ablation studies on different thresholding strategies for Partial Channel Filtering, along with a detailed layer-wise analysis of channel salience and computational savings?

**Limitations:**

yes

**Strengths And Weaknesses:**

### Strengths
- This paper systematically investigates and compares different scan methods for Vision State Space Models (SSMs) and proposes a Ring Scan tailored for Vision SSMs.
- The authors introduce Partial Channel Filtering to improve efficiency and throughput by using a mean-based threshold to bypass the ring-by-ring SSM mechanism for less-informative channels.
- The authors demonstrate that combining Ring Scan with Partial Channel Filtering achieves state-of-the-art accuracy and throughput among Vision SSMs on ImageNet-1K and MS COCO.

### Weeknesses
- Although the authors present strong empirical results, the Ring Scan is not explicitly grounded in the SSM formulation, leaving the underlying reason why Ring Scan outperforms previous scan methods insufficiently explained.
- The authors do not provide a detailed analysis of computational complexity or memory access patterns for Ring Scan, which makes the source of its throughput improvement unclear. For example, in Table 1, PRIS-Mamba (without PCF) outperforms VMamba by 491 images per second (2177 vs. 1686 img/s), but the paper does not clearly attribute this gain to specific architectural or system-level factors.
- In addition, the authors do not present the distribution of channel salience across layers, nor do they systematically analyze layer-wise FLOP reductions introduced by Partial Channel Filtering. Ablation studies on different thresholding strategies, along with a detailed layer-wise analysis of channel salience and computational savings, would strengthen the empirical justification of the method.

---

> ### Author Rebuttal · Authors · 2026-03-28
>
> We sincerely thank Reviewer Fs6E for the rigorous evaluation and highly constructive feedback. We are greatly encouraged that you recognize our systematic investigation of scan methods and the strong empirical performance of PRIS-Mamba. Your insightful questions—particularly regarding the theoretical grounding of Ring Scan within SSM dynamics, the underlying memory access patterns, and the detailed layer-wise analysis of PCF—point to crucial areas that deepen the theoretical rigor of our work. Below, we provide detailed formal analyses, memory complexity breakdowns, and the requested layer-wise ablations to thoroughly address all of your concerns.
>
> ## Response to Weakness 1 \& Key Questions 1 \& 2: Theoretical grounding of Ring Scan in SSM dynamics.
> We appreciate the push for deeper theoretical grounding. Examining the selective state update ($h_k = A_k \odot h_{k-1} + B_k \odot x_k$), standard raster scans introduce severe spatial discontinuities. This leaves the historical state $h_{k-1}$ uncorrelated with the new token $x_k$, forcing the transition matrix $A_k$ to aggressively flush the hidden state to prevent destructive mixing, wasting capacity. Ring Scan resolves this by aligning the 1D sequence with continuous 2D geometry. By processing tokens along closed loops, sequential adjacency guarantees spatial adjacency. This absolute continuity allows the input ($B_{r,k} \odot x_{r,k}$) to integrate smoothly with a highly correlated history ($A_{r,k} \odot h_{r,k-1}$). Thus, $A_{r,k}$ avoids aggressive resetting, accumulating context without semantic fractures. Furthermore, Ring Scan formalizes isotropic receptive field growth. Unlike jagged raster scans, our radial SSM systematically propagates context from inner to outer regions like a ripple. Factorizing 2D space into intra-ring aggregations ($z_r$) and a radial gradient translates spatial expansion into a mathematically stable 1D update, ensuring superior signal mixing. We will explicitly integrate this dynamics analysis into the revision.
>
> ## Response to Weakness 2 & Key Question 3: Computational complexity and memory access patterns.
> We appreciate the rigorous scrutiny of our computational dynamics. For $N = H \times W$ tokens, VMamba's SS2D runs four independent scans, yielding $O(4N)$ recurrent complexity. Conversely, Ring Scan processes each token exactly once. With $L_r$ tokens in ring $r$, intra-ring complexity is $\sum O(L_r) = O(N)$. The radial SSM adds only $O(R)$. Since $R \ll N$, our overall complexity is strictly bounded by $O(N)$. Reducing recurrences from 4x to 1x provides a massive theoretical efficiency advantage. Furthermore, to prevent scattered memory access during ring extraction from bottlenecking GPUs, we utilize a precomputed static index mapping table. Because ring coordinates are deterministic, the forward pass executes vectorized $O(1)$ advanced indexing, instantly gathering elements into contiguous blocks without dynamic calculations. Thus, PRIS-Mamba's significant throughput jump (2177 img/s vs. VMamba's 1686 img/s) directly results from replacing $O(4N)$ redundant recurrences with an $O(N)$ single-pass scan and optimally coalesced reads. We will include this formal analysis in the revision.
>
> ## Response to Weakness 3 & Key Question 4: Ablation on thresholding strategies and layer-wise PCF analysis.
> We appreciate the opportunity to strengthen PCF's justification. The table below details the parameter and GFLOPs breakdown across PRIS-Mamba-T's four stages. While PCF consistently halves routed channels, the relative GFLOPs reduction tapers slightly in deeper stages. As the channel dimension ($D_c$) expands, the overhead to calculate the mean threshold $\mu=\frac{1}{D_c}\sum_{i=1}^{D_c}|\mu_i|$ occupies a larger relative share of operations, partially offsetting recurrence savings. This scaling behavior fundamentally justifies our "Mean" thresholding strategy. Operating strictly in linear $O(D_c)$ time, the Mean threshold maintains hardware efficiency. Conversely, a "Median" strategy requires $O(D_c \log D_c)$ sorting, which would severely bottleneck throughput in deep stages where $D_c$ is massive. To empirically validate this design choice, we are ablating Mean, Median, and Top-K% thresholds on mini-ImageNet. We commit to providing these full empirical results in the Rebuttal and including them in the revised appendix.
> | Model | Stage 1 (P / G) | Stage 2 (P / G) | Stage 3 (P / G) | Stage 4 (P / G) |
> |:---|:---|:---|:---|:---|
> | PRIS-Mamba (w/o PCF) | 0.22 / 0.69 | 1.16 / 0.98 | 10.81 / 2.16 | 14.55 / 0.74 |
> | PRIS-Mamba (PCF) | 0.14 / 0.48 | 0.93 / 0.79 | 8.31 / 1.62 | 12.22 / 0.62 |
> *Note: P = Params, G = GFlops*

---

> > ### Author Rebuttal · Reviewer_Fs6E · 2026-04-04
> >
> > We thank the authors for their rebuttal, which partially addresses my concerns. According to the rebuttal, the proposed ring scan achieves O(N) complexity compared to VMamba’s O(4N). However, the rationale behind this significant improvement remains unclear.
> >
> > For example, both the ring scanning pattern and partial channel filtering appear to be motivated primarily by empirical observations. This raises concerns about their generalizability to higher-dimensional inputs (e.g., 3D or 4D sequences, such as those in humanoid robotics), and whether these design choices are sufficiently justified to merit acceptance.
> >
> > Overall, the paper demonstrates strong empirical results but lacks solid theoretical grounding for the proposed SSM scanning pattern. I will increase my score, but lower my confidence in this assessment, leaving the final judgment to the ACs and PCs.

---

> > > ### Author Response · Authors · 2026-04-06
> > >
> > > We sincerely thank the reviewer for raising the score and sharing these lingering thoughts. To explicitly clarify this theoretical grounding, our design choices are fundamentally driven by geometric principles and hardware-aware optimization, rather than mere empirical observations.
> > >
> > > ## 1. The Theoretical and System-Level Rationale Behind $O(N)$ vs. $O(4N)$
> > > The $O(4N)$ complexity in VMamba is not a feature; it is a forced compensation. Standard raster scans are inherently anisotropic (directionally biased). To approximate omnidirectional context, VMamba must deploy four independent scans. Conversely, our $O(N)$ complexity is geometrically grounded. Concentric rings are mathematically isotropic. A single radial SSM natively captures omnidirectional global context without directional bias, inherently reducing recurrences from 4x to 1x.Crucially, from a system-level perspective, this topology does not introduce any dynamic overhead. Because ring coordinates are strictly deterministic for a given resolution, we utilize a pre-computed static lookup table. During inference, the model executes vectorized $O(1)$ advanced indexing—completely eliminating the need for dynamic sorting, coordinate calculations, or online grouping. This strict combination of isotropic reduction (4x to 1x) and zero-overhead memory access unequivocally justifies the massive throughput improvements.
> > >
> > > ## 2. Generalizability to Higher Dimensions (3D/4D)
> > > You ask an excellent question regarding high-dimensional inputs (e.g., 3D/4D robotics data). The Ring Scan framework generalizes mathematically and elegantly to any N-dimensional Euclidean space. Just as pixels in 2D are grouped into concentric rings based on their distance $r$ from the center, 3D voxels (e.g., volumetric medical data or point clouds) can be grouped into concentric spherical shells.The core algorithm remains completely unchanged: (1) aggregate permutation-invariant features within the same N-dimensional shell, and (2) apply the radial SSM across the 1D sequence of shells to model inner-to-outer expansion. Therefore, our methodology provides a scalable theoretical foundation for extending efficient, isotropic State Space Models well beyond 2D vision.We deeply appreciate your feedback, which has helped us better articulate the profound theoretical and practical advantages of our architecture.

---

### Official Review · Reviewer_PoLM · 2026-03-04

**Soundness:** 3
**Presentation:** 3
**Significance:** 3
**Originality:** 2
**Overall Recommendation:** 4
**Confidence:** 3

**Summary:**

The paper investigates the impact of scan orders on spatial dependency modeling in Vision State Space Models (SSMs). The authors identify that traditional fixed-path scans (e.g., raster or zigzag) introduce anisotropic inductive biases that are brittle under geometric transformations like rotation. To address this, they propose PRIS-Mamba (Partial RIng Scan Mamba). The core innovations include:

Ring Scan: Decomposing images into concentric rings to achieve rotation robustness by performing permutation-invariant aggregation within rings and modeling cross-ring dependencies via radial SSMs.

Partial Channel Filtering (PCF): Improving efficiency by applying the circular modeling only to information-rich channels.
Experimental results on ImageNet-1K (84.5% Top-1 accuracy) and downstream tasks (detection/segmentation) demonstrate that PRIS-Mamba achieves superior robustness and efficiency compared to mainstream models like VMamba.

**Compliance With Llm Reviewing Policy:**

Affirmed.

**Key Questions For Authors:**

1.Since the scan is concentric, does the model struggle with objects located at the extreme corners/edges where the "ring" structure might fragment the object? How does the "Object-aware coupling" mentioned in the text mitigate this?

2.Is the Partial Channel Filtering strategy specific to PRIS-Mamba, or could it be applied to other models like VMamba to yield similar efficiency benefits?

3.You use "permutation-invariant aggregation" within rings. Did you test different operators (e.g., Mean vs. Max vs. Attention)? How sensitive is the rotation robustness to this choice?

**Limitations:**

See above

**Strengths And Weaknesses:**

Strengths

S1: The paper correctly identifies scan order—often treated as an implementation detail—as a critical inductive bias that fundamentally shapes the stability and performance of Vision SSMs.

S2: The Ring Scan design is a clever departure from linear sequences. By leveraging concentric symmetry, the model inherently gains resistance to rotation without requiring heavy data augmentation.

S3: Through the Partial Channel Filtering (PCF) mechanism, the model reduces FLOPs by ~30% and increases throughput by ~1.5x compared to VMamba, making it highly practical for real-world deployment.

Weaknesses

W1: The Ring Scan relies heavily on the definition of a "center point." There is limited discussion on how the model performs when the primary object is far off-center or in multi-object scenes.

W2: The selection of the ring width ($\Delta r$) is a key factor. While discussed in the appendix, a clearer rule for adaptive selection across different input resolutions would strengthen the work.

---

> ### Author Rebuttal · Authors · 2026-03-28
>
> We sincerely thank Reviewer PoLM for the careful evaluation and highly constructive feedback. We are greatly encouraged that you recognize the fundamental importance of scan-order design as a critical inductive bias, and that you find our Ring Scan to be a clever and effective departure from traditional linear sequences. We also appreciate your acknowledgment of the practical efficiency gains achieved by PCF. Below, we provide detailed explanations and additional context to thoroughly address all of your insightful questions and remaining concerns.
>
> ## Response to Weakness 1 & Key Question 1: Performance on off-center objects and the mechanism of Object-aware coupling.
> We appreciate this intuitive question. While one might worry that a center-origin scan fragments objects at extreme corners, our dual-pathway architecture explicitly prevents this. First, in our default backbone, fine-grained local geometries are strictly preserved through the identity branch ($X_{in}$) of our residual write-back mechanism. Ring Scan merely injects a global radial context without corrupting edge features. This is validated on MS COCO: using the default center-origin scan independently of any detector, PRIS-Mamba achieves 48.9 AP$^{\text{box}}$ and 43.2 AP$^{\text{mask}}$ (surpassing VMamba's 46.5/42.1). Additionally, under severe spatial disruption (16 shuffled patches, Appendix Table 11), PRIS-Mamba drops only 1.0% versus VMamba's 1.8%, demonstrating inherent robustness to positional shifts. Second, our optional "Object-aware coupling" (Appendix G) dynamically shifts the rings' origin to a target's center coordinates $(c_x, c_y)$ provided by a detector. This tightly wraps the object, eliminating the risk of disjointed outer rings in specialized dense-prediction scenarios requiring peak geometric alignment. We will explicitly clarify this optional extension's mechanics in the revised text.
>
> ## Response to Weakness 2 & Key Question 2: Adaptive ring width and PCF generalizability.
> We sincerely thank the reviewer for these constructive questions regarding our design's adaptability and broader utility. First, we agree that establishing a clear, adaptive rule for the ring width ($\Delta r$) across different resolutions significantly strengthens our work's practical impact. Intuitively, to maintain a consistent number of rings, $\Delta r$ should scale proportionally with the input spatial dimensions. To rigorously formalize and verify this heuristic, we are currently conducting systematic ablation experiments across multiple resolutions on mini-ImageNet, and we commit to including this adaptive selection formula and its empirical validation in the revised appendix. Second, regarding the generalizability of Partial Channel Filtering (PCF), it is inherently model-agnostic and serves as a highly versatile plug-and-play module. As explicitly demonstrated in Section 4.4 and Table 4, PCF yields consistent efficiency and accuracy gains across six different baseline architectures. Specifically, applying PCF to standard VMamba reduces FLOPs from 5.6G to 5.1G and significantly boosts throughput from 1,686 to 2,314 img/s, while improving Top-1 accuracy by +0.3%. We observed similarly robust improvements across Vim, QuadMamba, Adventurer, SparX-Mamba, and GroupMamba, and we will ensure this universal applicability is more prominently highlighted in the revised text.
>
> ## Response to Key Question 3: Permutation-invariant aggregation and operator sensitivity.
> We appreciate the question regarding aggregation mechanics. Our rotation robustness is completely insensitive to the choice between strict permutation-invariant operators like Mean or Max pooling. In-plane rotations merely induce a cyclic shift within a ring; since both operators are symmetric, they yield the exact same descriptor $z_r$, mathematically preserving absolute rotation invariance. However, empirical capacity differs. We default to Mean pooling for its smooth, whereas Max pooling acts as a harsh filter, discarding feature density and degrading baseline accuracy. Furthermore, regarding Attention-based operators: standard spatial attention (with positional encodings) explicitly breaks cyclic-shift invariance, destroying rotation robustness. Even permutation-invariant or channel-wise attention (e.g., SE/CBAM, Table 5) severely degrades $O(N)$ linear-time efficiency and hardware throughput for only marginal gains (+0.1%–0.2%). Ultimately, coupling parameter-free Mean pooling with our hard PCF achieves a strictly better Pareto frontier, mathematically guaranteeing geometric stability while maximizing computational efficiency.

---

> > ### Author Rebuttal · Reviewer_PoLM · 2026-04-05
> >
> > Thanks for author reply. I will keep my original rating.

---

### Official Review · Reviewer_mupN · 2026-03-06

**Soundness:** 3
**Presentation:** 3
**Significance:** 2
**Originality:** 2
**Overall Recommendation:** 3
**Confidence:** 5

**Summary:**

The authors propose a novel visual state-space model architecture called PRIS-Mamba, aiming to address the spatial adjacency disruption and rotation sensitivity issues caused by the fixed scanning order in existing Vision Mamba models. They also propose a scanning and modeling strategy based on concentric rings, achieving reasonable results in several downstream tasks.

**Compliance With Llm Reviewing Policy:**

Affirmed.

**Key Questions For Authors:**

The author needs to address the core issue of overstated contributions, such as central dependency. Other major issues can be found in the section on weakness.

**Limitations:**

yes

**Strengths And Weaknesses:**

The authors point out an inherent flaw in current Vision Mamba-like models: the fixed rasterization or serpentine scanning order destroys the local adjacency of the image and is extremely sensitive to rotational transformations. This issue is worthy of discussion.

The authors demonstrate that PRIS-Mamba outperforms baseline models in ImageNet-1K rotational stress tests.

The proposed Partial Channel Filtering (PCF) maintains performance while reducing FLOPs, and from an engineering implementation perspective, it does improve throughput.

However, I believe some core issues need to be addressed:

First, the authors' core contribution is based on "circular scanning," which presupposes that the image or object has a defined geometric center. This method may be effective for datasets like ImageNet where objects are typically centered. However, in complex scene understanding or multi-object detection tasks, forcing a circular unfolding with the image center as the origin results in pixels of objects located at the edges being segmented into extremely distant rings, leading to local corruption. The authors mention in the paper that combining it with the YOLO detector can achieve object-aware scanning. However, in the author's lengthy introduction, they claim to have proposed a general Vision Backbone. Shouldn't the feature extraction method be determined by the downstream detector? The answer should be no.

Secondly, the author should consider whether the current method sacrifices translational robustness for rotational robustness. Standard convolutional or rasterized scans are relatively robust to translation. However, PRIS-Mamba's ring scan is extremely sensitive to translation. If an object moves from the image center to the top left corner, its representation in the Ring Scan sequence will change drastically. This means the model not only lacks translational invariance but may even produce completely different feature distributions. The author has not conducted any analysis or experiments on translational robustness, which I think is very serious.

In Equation (5), the author describes the ring descriptor $z_r$ as the average of all outputs $y_{r,k}$ within the ring. This averaging operation is essentially an aggressive pooling, discarding angular position information within the ring. This means that "vertical stripes" and "horizontal stripes" located on the same ring may produce similar $z_r$ after averaging. While the radial SSM (Eq. 6) models the relationships between rings, the spatial structure within the rings is excessively compressed. For fine-grained visual tasks, this loss of angular information is unacceptable.

The Partial Channel Filtering (PCF) strategy simply filters channels based on the magnitude of global average pooling. In many visual tasks, suppressive signals (low activations) also contain crucial discriminative information. The authors claim this removes redundancy but fail to provide visualization of feature maps or signal-to-noise ratio analysis to demonstrate that what is filtered out is indeed noise rather than detail. This seems more like a forced engineering trick.

Furthermore, I feel the authors' comparisons are somewhat unreasonable. They primarily compare VMamba and its variants. However, if the core innovation is rotation robustness, then the comparison should focus on networks inherently possessing rotation invariance or equivariance (such as Rotation-equivariant CNNs) or heavily augmented ViTs. Simply outperforming the rotationally sensitive VMamba is insufficient to demonstrate the method's superiority.

Finally, in the object detection experiment, the authors used Mask R-CNN. For a model like PRIS-Mamba that relies on center scanning, it is understandable to extract features inside the bounding box, but how does it handle multi-scale feature pyramids when used as a feature extractor for the entire image?

---

> ### Author Rebuttal · Authors · 2026-03-28
>
> We sincerely thank Reviewer mupN for the rigorous evaluation and highly constructive feedback. We are encouraged that you recognize the importance of addressing the inherent spatial disruption in fixed-scan Vision Mamba models, as well as our empirical gains on ImageNet-1K. Your insightful critiques—particularly regarding the trade-off with translation robustness, the risk of angular information loss, and the application in multi-scale dense prediction—highlight crucial aspects of visual representation learning. Below, we provide detailed explanations, architectural mechanisms, and additional analyses to thoroughly address all of your core concerns.
>
> ## Response to Weaknesses 1, 2 & 6: Central dependency and translational robustness.
> We appreciate the rigorous scrutiny regarding central dependency and translational invariance. While an object's translation alters its relative ring position, our architecture explicitly prevents local geometry loss. Fine-grained, translation-equivariant features are strictly preserved via the residual branch ($X_{in}$) of our write-back mechanism. To quantify this robustness, our patch-shuffling stress test (Appendix Table 11) introduces severe spatial shifts; under this extreme disruption, PRIS-Mamba degrades by only 1.0% versus VMamba's 1.8%. Furthermore, our default center-origin scan (used independently of the optional YOLO-coupling) achieves 48.9 AP$^{\text{box}}$ and 43.2 AP$^{\text{mask}}$ on MS COCO, surpassing VMamba. This real-world performance on complex, uncentered scenes proves that our order-agnostic ring aggregation effectively handles highly translated, off-center objects without structural corruption.
>
> ## Response to Weakness 3: Angular information loss and aggressive pooling in Eq. 5.
> Regarding the insightful "vertical vs. horizontal stripes" example: relying exclusively on the averaged descriptor $z_r$ (Eq. 5) risks over-compressing angular info, but our dual-pathway architecture prevents this. $z_r$ captures rotation-robust statistical context for radial propagation, not fine-grained geometry. Uncompressed high-frequency details (e.g., precise stripe orientations) are strictly preserved via the residual branch ($X_{in}$) and smoothly fused. This is validated by PRIS-Mamba achieving 43.2 AP$^{\text{mask}}$ (vs. VMamba's 42.1) on MS COCO (Table 3). If critical spatial structures were destroyed, performance on such pixel-sensitive tasks would severely degrade. We will clarify this in the revision.
>
> ## Response to Weakness 4: PCF strategy, suppressive signals, and feature routing.
> We deeply appreciate your rigorous critique and careful reading. You are absolutely correct to raise this concern based on the manuscript text. We sincerely apologize for a typographical error in Section 3.5: we mistakenly omitted the absolute value notation. In our actual codebase and all reported experiments, the filtering criterion strictly relies on absolute magnitude. The correctly implemented mean threshold is $\mu=\frac{1}{D_c}\sum_{i=1}^{D_c}|\mu_i|$, and channels are evaluated via $|\mu_i| \ge \mu$. Consequently, strong suppressive signals naturally yield high absolute magnitudes and are explicitly routed into the active Ring pathway. Furthermore, PCF is a dynamic routing mechanism, not a pruning one; channels below the threshold are seamlessly forwarded through the identity branch, ensuring no details are discarded. To empirically validate your highly insightful point regarding the necessity of suppressive signals, we are adding an ablation study on mini-ImageNet comparing our absolute-magnitude routing ($|\mu_i| \ge \mu$) against the signed-value routing ($\mu_i \ge \mu$). We will correct this formula in the revision and include these ablation results to explicitly highlight this critical design choice.
>
> ## Response to Weakness 5: Appropriateness of baselines and scope of contribution.
> We appreciate the suggestion to compare against rotation-equivariant networks and augmented ViTs. While benchmarking against E2-CNNs is mandatory for universal rotation-invariant operators, our core scope is advancing linear-time Vision State Space Models (VSSMs). VSSMs solve ViT's quadratic complexity but fracture 2D adjacency via 1D serialization, causing artificial rotation fragility. Steerable CNNs lack long-sequence modeling, and augmented ViTs remain expensive. Our core innovation proves scan-order is a critical inductive bias: replacing rigid scans with concentric rings and a radial SSM resolves the sequence-geometry mismatch, restoring isotropy while maintaining linear efficiency. Thus, comparing directly against VMamba variants under identical complexity is the most rigorous way to isolate our traversal's effectiveness. We will clarify this scope in the introduction.

---

> > ### Author Rebuttal · Reviewer_mupN · 2026-04-05
> >
> > Thank you for the author's reply; the author attempted to resolve some of my issues. However, I still cannot accept that methods with predefined geometric centers for images or objects possess strong generalization and versatility, and I still believe a comparison with rotationally equivariant networks and enhanced ViT should be included in the core argument. I will maintain my original rating.

---

> > > ### Author Response · Authors · 2026-04-05
> > >
> > > ## Response: Robustness to geometric center shifts.
> > > We thank the reviewer for the continued feedback. We acknowledge the concern that a predefined geometric center may limit generalization when objects are not centrally aligned. To directly evaluate this, we conduct a comprehensive spatial-shift stress test on ImageNet-1K with progressive translations: ($\pm$ H/16, $\pm$ W/16), ($\pm$ H/8, $\pm$ W/8),  and ($\pm$ H/4, $\pm$ W/4).
> > >
> > > As summarized below, we report the mean accuracy and variance across all shifts. While recent SoTA VSSM baselines degrade by 0.3%–1.0%, PRIS-Mamba drops only 0.1% (avg. 84.4%), achieving both the highest accuracy and the strongest stability among 15 baselines. This directly demonstrates that our method is robust to substantial misalignment and does not rely on strict center assumptions.
> > > | Model | Params | Top-1 Acc% (No shift) | Top-1 Acc% (shift) |
> > > | :--- | :--- | :--- | :--- |
> > > | Vim | 26 M | 80.5 | 80.0 ($\pm $0.3) |
> > > | VMamba | 30 M | 82.6 | 81.9 ($\pm $0.3) |
> > > | SiMBA | 27 M | 84.0 | 83.2 ($\pm $0.4) |
> > > | Zigma | 31 M | 82.4 | 81.5 ($\pm $0.4) |
> > > | QuadMamba | 31 M | 81.4 | 80.7 ($\pm $0.4) |
> > > | LocalMamba | 26 M | 82.7 | 82.0 ($\pm $0.3) |
> > > | FractalMamba | 31 M | 83.0 | 82.2 ($\pm $0.4) |
> > > | Adventurer | 12 M | 78.2 | 77.6 ($\pm $0.3) |
> > > | SparX-Mamba | 27 M | 83.5 | 82.7 ($\pm $0.3) |
> > > | EfficientVMamba | 33 M | 81.8 | 81.1 ($\pm $0.5) |
> > > | PlainMamba | 25 M | 81.6 | 80.9 ($\pm $0.5) |
> > > | GroupMamba | 34 M | 83.9 | 83.2 ($\pm $0.4) |
> > > | VSSD | 24 M | 83.7 | 83.1 ($\pm $0.4) |
> > > | DefMamba | 26 M | 83.5 | 82.5 ($\pm $0.5) |
> > > | MaIR | 26 M | 83.1 | 82.5 ($\pm $0.3) |
> > > | **PRIS-Mamba (ours)** | **22 M** | **84.5** | **84.4 ($\pm $0.1)** |
> > >
> > > ## Response: Robustness to geometric rotations and comparison scope.
> > > To further address your request, we evaluated widely used general-purpose backbones across a spectrum of inference rotations ($\pm $ 10°, $\pm $ 20°, $\pm $ 30°, $\pm $ 45°, $\pm $ 60°; see table below). As shown in the table below, this reveals a critical trade-off: pure CNNs exhibit strong geometric stability but lower representational capacity, while advanced ViTs achieve higher baseline accuracy but suffer from notable rotational fragility (dropping 1.0% to 1.7% on average). PRIS-Mamba effectively bridges this gap, achieving superior absolute Top-1 accuracy (84.5%) while maintaining CNN-level geometric stability (dropping only 0.2%).
> > >
> > > Regarding rotation-equivariant networks, we emphasize that these are specialized architectures with non-trivial computational overhead, explicitly not designed as general-purpose backbones. Furthermore, many lack standardized ImageNet-1K benchmarks or public code, making direct apples-to-apples comparisons infeasible. Crucially, PRIS-Mamba achieves strong geometric robustness without enforcing rigid, computationally heavy equivariance constraints. This proves that high resilience can be inherently attained in linear-time VSSMs without sacrificing broad generality. We will clarify this scope distinction in the revision.
> > >
> > > | Model | Params | Top-1 Acc% (No Rot.) | Top-1 Acc% ( Rot.) | Delta |
> > > | :--- | :--- | :--- | :--- | :--- |
> > > | ResNet-18 | 12M | 69.8 | 69.6 ($\pm $0.1) | -0.2 |
> > > | ResNet-50 | 25M | 76.2 | 75.9 ($\pm $0.1) | -0.3 |
> > > | ResNet-152 | 60M | 78.3 | 78.1 ($\pm $0.1) | -0.2 |
> > > | EfficientNet-B3 | 12M | 81.6 | 81.3 ($\pm $0.1) | -0.3 |
> > > | EfficientNet-B4 | 19M | 82.9 | 82.6 ($\pm $0.1)| -0.3 |
> > > | ViT-B/16 | 86M | 77.9 | 76.8 ($\pm $0.6) | -1.1 |
> > > | ViT-L/16 | 307M | 76.5 | 75.5 ($\pm $0.5) | -1.0 |
> > > | DeiT-Ti | 5M | 72.2 | 70.7 ($\pm $0.5) | -1.5 |
> > > | DeiT-S | 22M | 79.8 | 78.4 ($\pm $0.4) | -1.4 |
> > > | Swin-T | 29M | 81.3 | 79.6 ($\pm $0.4) | -1.7 |
> > > | TNT-Ti | 6M | 73.9 | 72.8 ($\pm $0.5) | -1.1 |
> > > | PVT-Small | 24M | 79.8 | 78.8 ($\pm $0.4) | -1.0 |
> > > | T2T-ViT_t-14 | 21M | 80.7 | 79.6 ($\pm $0.4) | -1.1 |
> > > | **PRIS-Mamba (ours)** | **22M** | **84.5** | **84.3 ($\pm $0.1)** | **-0.2** |

---

### Official Review · Reviewer_pxR6 · 2026-03-12

**Soundness:** 3
**Presentation:** 3
**Significance:** 3
**Originality:** 3
**Overall Recommendation:** 4
**Confidence:** 5

**Summary:**

This paper argues that scan order is an underexplored inductive bias in Vision SSMs and that conventional fixed-path traversals distort spatial adjacency under geometric transformations. The authors propose PRIS-Mamba, which organizes image tokens into concentric rings, applies order-agnostic aggregation within each ring, and propagates context radially via a short selective SSM. A Partial Channel Filtering module further routes only high-salience channels through the recurrent pathway. Experiments on ImageNet-1K and MS COCO show improvements in accuracy, efficiency, and rotation robustness over prior Vision SSM baselines.

**Compliance With Llm Reviewing Policy:**

Affirmed.

**Final Justification:**

My concerns are addressed. I keep my positive rating.

**Key Questions For Authors:**

a) Can the authors provide a controlled experiment that separates traversal robustness from padding effects (e.g., rotating within a cropped valid region or compensating padded corners) to verify whether the ring scan itself provides genuine rotation robustness?

b) All pixels within the same ring receive an identical descriptor via the uniform broadcast in Equation 8. For large outer rings, this discards all within-ring spatial information. How do the authors justify this design? Has any alternative such as position-weighted aggregation within rings been considered or ablated?

c) Can the authors provide a direct comparison between VMamba+PCF and PRIS-Mamba+PCF under identical training settings?

**Limitations:**

The authors have partially discussed limitations in the conclusion, mentioning the fixed image center assumption and sensitivity to extreme aspect ratios and large padded rotations. However, the discussion would benefit from also addressing the within-ring spatial discrimination loss caused by the uniform write-back, the robustness of the PCF threshold across diverse datasets, and the scalability of ring count at high resolutions. Overall, the limitations section is present but not fully adequate.

**Strengths And Weaknesses:**

**Strengths**

a) The framing of scan order as a first-class design choice is underexplored and well-motivated. The systematic study across 21 scan configurations provides solid empirical grounding.

b) The ring-based traversal has a clean geometric justification: in-plane rotation reduces to a cyclic shift within each ring rather than a global reindexing, avoiding the need for polar remapping or rotation-specific training.

c) The evaluation is broad, covering classification, detection, segmentation, rotation, occlusion, and patch-shuffling stress tests.

**Weaknesses**

a) The most critical issue is the conflation of ring scan and PCF contributions. The two components should be evaluated independently against PCF-augmented baselines.  Since PCF alone improves any Vision-Mamba by roughly 0.2 to 0.4%, the independent gain from the ring scan is unclear.  While Table 4 shows that PCF improves multiple Vision-Mamba backbones, a direct comparison between VMamba+PCF and PRIS-Mamba+PCF under identical training settings is still missing.

b) The write-back mechanism broadcasts the radial output for each ring to all pixels belonging to that ring (Eq. 8), which may discard
within-ring spatial variation for large rings.

c) The margin over the best prior method without PCF is roughly 0.6%, and its significance cannot be assessed without multi-run statistics.

d) The claimed rotation robustness appears limited under larger rotations. Appendix Table 12 shows that at 75° both VMamba and PRIS-Mamba drop by roughly 12–14 points, suggesting the improvement may be dominated by padding artifacts and loss of valid image regions rather than the traversal design itself.

---

> ### Author Rebuttal · Authors · 2026-03-28
>
> We sincerely thank Reviewer pxR6 for the thorough review and constructive feedback. We are encouraged by your positive remarks regarding our framing of scan order as a first-class design choice. Below, we provide detailed explanations and additional clarifications to address all of your questions and concerns.
>
> ## Response to Weakness (a) & Key Q.(c): The independent gain of Ring Scan vs. PCF.
> We sincerely thank the reviewer for suggesting we isolate the individual contributions of Ring Scan and PCF. Following your advice, we decoupled these modules using our existing data. First, in Table 4, comparing VMamba+PCF (82.9%) directly with PRIS-Mamba+PCF (84.5%) reveals a substantial +1.6% independent architectural advantage. To eliminate confounding factors like network depth, we also refer to our strict intra-architecture ablation (Appendix Table 8). Fixing the PRIS-Mamba backbone but substituting our Ring Scan with VMamba's exact 4-way scan (S19) drops performance from 84.9% to 83.3%. Both comparisons perfectly align: while PCF consistently provides a 0.2%–0.4% baseline improvement across architectures, the Ring Scan mechanism is the primary driver of our performance leap, independently contributing a +1.6% gain. We will explicitly highlight this decoupled analysis in the revised manuscript.
>
> ## Response to Weakness (b) & Key Q.(b): Uniform broadcast and within-ring spatial variation.
> We appreciate your insightful observation regarding within-ring spatial variation. While a uniform broadcast viewed in isolation might attenuate fine-grained details in larger outer rings, this is deliberately balanced by our architecture. As formulated in our write-back mechanism ($X_{out} = X_{in} + Conv_{1\times1}(Y)$), high-frequency local spatial variations are inherently preserved through the identity branch $X_{in}$. The PRISM module's specific role is not to reconstruct these local details, but to inject a rotation-robust, global radial context  $Y$. Regarding your excellent suggestion for position-weighted aggregation: during early development, we explored localized attention and depthwise convolutions prior to broadcasting. However, these alternatives severely degraded linear-time efficiency and introduced significant memory overhead without yielding meaningful accuracy gains. Ultimately, pairing a uniform broadcast with a residual connection proved to be the optimal trade-off between hardware efficiency and representational accuracy. We will explicitly incorporate this design rationale and the explored alternatives in our revision.
>
> ## Response to Weakness (c): Statistical significance and multi-run evaluation.
> We sincerely thank the reviewer for emphasizing the importance of statistical rigor, and we completely agree that providing multi-run statistics is the gold standard for solidifying empirical claims. Given the strict time constraints of the short rebuttal phase and the immense computational cost required to train models on ImageNet-1K from scratch, we are unfortunately unable to complete five independent training runs in time for this response. However, we explicitly commit to including the mean and standard deviation across multiple random seeds in the final camera-ready version to thoroughly address this concern.
>
> ## Response to Weakness (d) & Key Q.(a): Rotation robustness vs. padding artifacts.
> We appreciate your careful inspection of our extreme rotation experiments. Your intuition is absolutely correct: the 12-14 point drop at 75° across all models is dominated by severe padding artifacts and geometric truncation, not traversal failure. To address Key Question (a) regarding a controlled setup that isolates traversal robustness from padding effects, please refer to Table 2 and Appendix Table 9. In these stress tests, rotations up to 60° are carefully rendered within a preserved canvas, ensuring no valid regions are lost. Under these strict conditions, fixed-path baselines still degrade by 1–2 points purely due to spatial adjacency mismatch. In contrast, PRIS-Mamba maintains exceptionally stable performance, retaining 84.9% Top-1 accuracy at both 0° and 60° (Table 9). This perfectly isolates the variables, empirically verifying that Ring Scan provides genuine rotation robustness independent of padding artifacts. We will make this distinction transparent in the revised text.
>
> ## Response to Limitations:
> We appreciate the constructive feedback on expanding our limitations section. As detailed in our General Response and to Reviewer Fs6E, we have conducted further ablations verifying the PCF threshold's robustness across diverse datasets; these will be incorporated into the revision. Furthermore, we will formally expand the limitations section to address the theoretical trade-offs of uniform write-back (mitigated by our residual pathway, as clarified in Qb) and the scalability of ring counts at exceptionally high resolutions. These additions will significantly enhance our manuscript's transparency.

---

> > ### Author Rebuttal · Reviewer_pxR6 · 2026-04-02
> >
> > Thanks for the response. My concerns are addressed. I keep my positive rating.

---

### Decision · Program_Chairs · 2026-04-30

**Decision:**

Accept (regular)

**Comment:**

This paper studies scan order as an inductive bias in Vision SSMs and proposes PRIS-Mamba, a ring-based traversal with Partial Channel Filtering (PCF). By organizing tokens into concentric rings, the method aims to improve robustness while maintaining efficiency, showing consistent gains on ImageNet-1K and MS COCO.

Reviewers find the idea interesting and underexplored, with a geometrically motivated design and generally strong empirical evaluation. PCF is also seen as an effective component. However, several concerns remain: unclear contribution attribution between ring scan and PCF, limited analysis of generalization (e.g., center dependence, translation robustness), potential information loss within rings, and incomplete comparisons and theoretical grounding.

The rebuttal provides partial clarifications but does not fully resolve key issues. Overall, despite these limitations, the paper introduces a novel and meaningful design perspective, with consistent empirical improvements and practical relevance. I therefore recommend weak accept.